# Divide and Merge: Motion and Semantic Learning in End-to-End Autonomous Driving

Yinzhe Shen[1]    Ömer Şahin Taş[1,2]    Kaiwen Wang[1]    Royden Wagner[1]    Christoph Stiller[1,2]

[1]Karlsruhe Institute of Technology (KIT)
[2]FZI Research Center for Information Technology

Reviewed on OpenReview: https://openreview.net/forum?id=RvtCNm1Rdv

## Abstract

Perceiving the environment and its changes over time corresponds to two fundamental yet heterogeneous types of information: semantics and motion. Previous end-to-end autonomous driving works represent both types of information in a single feature vector. However, including motion related tasks, such as prediction and planning, impairs detection and tracking performance, a phenomenon known as negative transfer in multi-task learning. To address this issue, we propose Neural-Bayes motion decoding, a novel parallel detection, tracking, and prediction method that separates semantic and motion learning. Specifically, we employ a set of learned motion queries that operate in parallel with detection and tracking queries, sharing a unified set of recursively updated reference points. Moreover, we employ interactive semantic decoding to enhance information exchange in semantic tasks, promoting positive transfer. Experiments on the nuScenes dataset with UniAD and SparseDrive confirm the effectiveness of our divide and merge approach, resulting in performance improvements across perception, prediction, and planning. Our code is available.

## 1 Introduction

Modular end-to-end (E2E) autonomous driving (AD) is gaining attention for combining the strengths of traditional pipeline methods with strict E2E approaches. In this framework, perception, prediction, and planning form the core set of tasks, which ideally complement one another to enhance overall system performance, presenting a multi-task learning challenge. However, a poorly designed multi-task learning structure could not only fail to facilitate mutual learning but also adversely affect individual tasks, a phenomenon known as negative transfer (Crawshaw, 2020). The prevalent modular E2E approaches (Hu et al., 2023; Jiang et al., 2023; Zheng et al., 2025; Sun et al., 2024) typically employ a sequential structure (Fig. 1a), where object embeddings are shared for learning the appearance and motion information. Although the sequential structure aligns with how humans perform driving tasks and has demonstrated promising planning performance, approaches using this structure exhibit negative transfer in object detection and tracking. In other words, the perception performance of jointly trained E2E models is typically inferior to those trained without the motion prediction and planning tasks, potentially leading to suboptimal final planning performance.

We analyze the underlying causes of negative transfer by inspecting the types of learned heterogeneous information: semantic and motion. Semantic information encompasses the categories of surrounding objects, lanes, crossings, *etc.*, while motion information describes the temporal changes occurring within the environment. Sequential methods (Hu et al., 2023; Jiang et al., 2023; Zheng et al., 2025; Doll et al., 2024) execute these two processes in succession. They first conduct detection and tracking and then use the extracted object features for trajectory prediction. This sequential design forces the features to contain motion information, compromising the initially learned semantic and leading to negative transfer in perception. The SHAP values

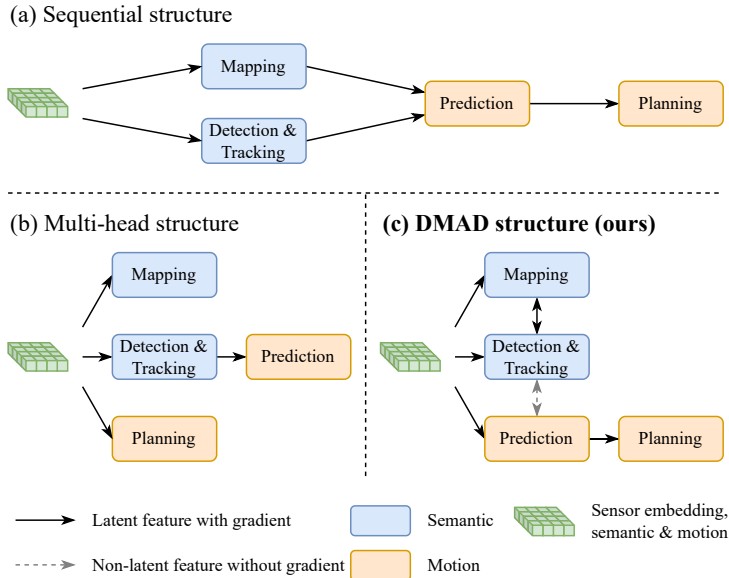

Figure 1: **Comparison of E2E structures.** In (a), semantic and motion learning occur sequentially. In (b), the multi-head structure parallelizes tasks with different heads; however, motion and semantic learning remain sequential in detection, tracking, and prediction. In (c), semantic and motion learning are performed in parallel without latent feature sharing or gradient propagation. In contrast, the exchange of information between the object and map perception modules is enhanced.

analysis (Lundberg & Lee, 2017) provides supporting evidence for our argument. Another E2E structure is depicted in Fig. 1b. It executes most tasks with different heads in parallel, as in PARA-Drive (Weng et al., 2024) and NMP (Zeng et al., 2019). However, since detection and prediction remain sequential, the issue of negative transfer persists.

In this work, we propose **DMAD structure** (Fig. 1c), **D**ividing and **M**erging motion and semantic learning for E2E **A**utonomous **D**riving. DMAD addresses the issue of negative transfer by separating semantic and motion learning. Furthermore, it leverages correlations among semantic tasks by merging them.

For dividing, we propose **Neural-Bayes motion decoder**. We maintain a set of motion queries that attend to the sensor embeddings parallel to the object (detection and tracking) queries. The key difference between motion and object queries is that they are decoded into past and future trajectories rather than bounding boxes with classes. Motion and object queries share a single set of reference points, updated recursively by detection and prediction. It allows only limited information exchange between both types of queries, mediated through the reference points without gradient flow. Moreover, we calculate the object's velocity using the predicted trajectory with finite differences, thereby removing the requirement for object queries to learn the velocity directly. In this manner, the object query focuses on learning semantic and appearance features, while the motion query is dedicated to capturing motion features. The two types of heterogeneous information are learned separately along distinct paths, effectively preventing negative transfer. Notably, the DMAD structure promotes motion learning to the same level of semantic learning, treating detection, tracking, and prediction as concurrent tasks for the first time, to the best of our knowledge.

For merging, we propose **interactive semantic decoder** to enhance the exchange of semantic insights in detection and map segmentation. Object perception and map perception are inherently related tasks. Previous methods often overlook this connection, typically executing the two along parallel paths (Hu et al., 2023; Jiang et al., 2023; Zheng et al., 2025). DualAD (Doll et al., 2024) leverages this correlation but allows only object perception to learn from the map. Our method uses layer-wise iterative self-attention (Vaswani et al., 2017) to enable mutual learning between object and map tasks, fostering positive transfer.

Experiments on the nuScenes (Caesar et al., 2020) dataset showcase the effectiveness of DMAD structure in mitigating negative transfer. Our approach achieves significant performance gains in perception and prediction, which benefits the planning module and outperforms state-of-the-art (SOTA) E2E AD models.

Our key contributions are summarized as follows:

- We examine the similarity and heterogeneity among tasks in modular E2E AD and argue that the prevailing design—learning information for conflicting tasks such as detection and prediction within a single feature—causes negative transfer in perception. We analyze SHAP values to validate this hypothesis. Conversely, we propose that information exchange between similar tasks, like detection and mapping, can facilitate positive transfer.

- We propose DMAD, a modular E2E AD paradigm that divides and merges tasks according to the information they are supposed to learn. This design eliminates negative transfer between different types of tasks while reinforcing positive transfer among similar tasks.

- We introduce two decoders: the Neural-Bayes motion decoder for concurrent trajectory prediction with object detection and tracking; the interactive semantic decoder to enhance information sharing between object and map perception. The proposed decoders improve existing SOTA methods, leading to better performance across all tasks.

## 2 Related Work

**Semantic learning.** Semantic learning includes object detection and map segmentation. Multi-view cameras have become popular due to their cost-effectiveness and strong capability in capturing semantic information. Current SOTA object detection and mapping approaches are built on the DETR (Carion et al., 2020) architecture, using a set of queries to extract semantic information from environment features through cross-attention (Vaswani et al., 2017) mechanisms. Sparse methods (Wang et al., 2022; Lin et al., 2022) learn semantic information by projecting queries onto the corresponding image features, focusing on the relevant regions. The PETR series (Liu et al., 2022; 2023; Wang et al., 2023) embed 3D positional encoding directly into 2D image features, eliminating the need for query projection. Another line of work aggregates all image features into a bird's-eye view (BEV) feature (Philion & Fidler, 2020; Li et al., 2022; Yang et al., 2023; Pan et al., 2024; Liao et al., 2023; 2024). Propagating the object queries over time enables multi-object tracking (Zeng et al., 2022; Meinhardt et al., 2022). This same technique is also used in map perception (Chen et al., 2025). Although tracking is also a motion-related task, we classify it as a semantic task, as query-based trackers learn only velocities as the motion information, as elaborated in Appendix B.

**Motion learning.** By motion, we refer to trajectory prediction and planning. Trajectory prediction studies typically use the ground truth of objects' historical trajectories along with high-definition maps as inputs. Early approaches (Chai et al., 2019; Cui et al., 2019; Bansal et al., 2019) rasterize maps and trajectories into a BEV image, using CNNs to extract scene features. Vectorized methods (Gao et al., 2020; Zhou et al., 2022) represent elements using polygons and polylines, using GNNs or Transformers to encode the scene (Ngiam et al., 2022; Wagner et al., 2024; Shi et al., 2022; Gu et al., 2021; Zhang et al., 2024).

For planning, imitation learning is a straightforward approach to E2E planning, where a neural network is trained to plan future trajectories or control signals directly from sensor data, minimizing the distance between the planned path and the expert driving policy (Bojarski, 2016; Prakash et al., 2021; Chen & Krähenbühl, 2022). Many approaches incorporate semantic tasks as auxiliary components to support E2E planning, using the nuScenes (Caesar et al., 2020) dataset and open-loop evaluation. These methods go beyond pure motion learning and are presented in the next paragraph. AD-MLP (Zhai et al., 2023) and Ego-MLP (Li et al., 2024) utilize only the ego vehicle's past motion states and surpass methods that rely on sensor inputs in open-loop evaluation. It aligns with our argument that semantics and motion are heterogeneous: AD-MLP and Ego-MLP can concentrate on learning from expert motion data without interference by irrelevant semantic information, thereby achieving superior open-loop planning performance.

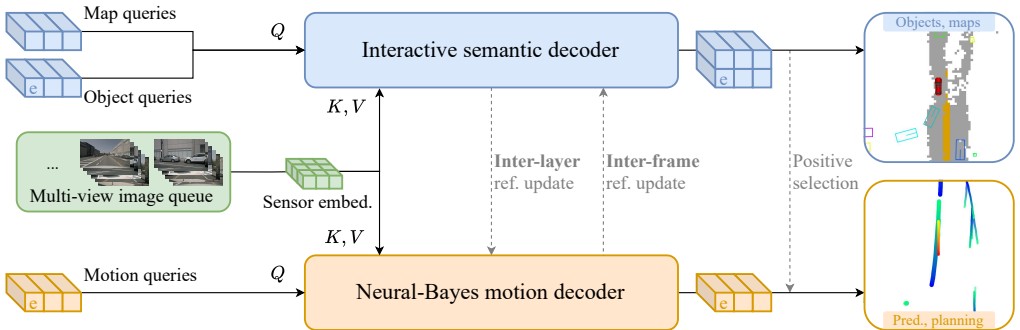

Figure 2: **An overview of DMAD.** A backbone processes multi-view images into sensor embeddings. Map and object queries are initialized, then interactively attend to the sensor embeddings for map and object perception. Motion queries, mapped one-to-one with object queries, share reference points that are iteratively updated. Finally, motion queries corresponding to detected objects are decoded into future trajectories. The ego motion query ("e") is used for planning. Gray dashed lines indicate operations without gradient flow.

**Joint semantic and motion learning.** E2E perception and prediction approaches learn semantics and motion jointly. The pioneering work FaF (Luo et al., 2018) uses a prediction head, in addition to the detection head, to decode the object features into future trajectories. Some works (Casas et al., 2018; Djuric et al., 2021; Fadadu et al., 2022) enhance it with intention-based prediction and refinement. PnPNet (Liang et al., 2020) and PTP (Weng et al., 2021) involve tracking, *i.e.*, jointly optimizing detection, association, and prediction tasks. While PTP performs tracking and prediction in parallel, it cannot predict newly emerging objects due to the lack of concurrent detection—a limitation our method successfully overcomes. ViP3D (Gu et al., 2023) first extends the query-based detection and tracking framework (Zeng et al., 2022) to prediction. Each query represents an object and propagates across frames. In each frame, queries are decoded into bounding boxes and trajectories using high-definition maps as additional context.

To include planning, NMP (Zeng et al., 2019) extends IntentNet (Casas et al., 2018) with a sampling-based planning module, where prediction is leveraged to minimize collisions during the planning process. Other works (Chitta et al., 2021; Casas et al., 2021; Hu et al., 2022) incorporate map perception as an auxiliary task. With the growing popularity of query-based object detectors (Carion et al., 2020; Li et al., 2022) and trackers (Zeng et al., 2022; Meinhardt et al., 2022), recent modular E2E AD approaches represent objects as queries, similar to ViP3D (Gu et al., 2023). UniAD (Hu et al., 2023) and its variants (Doll et al., 2024; Weng et al., 2024) retain the query propagation mechanism for tracking, aiming to explicitly model objects' historical motion. In contrast, VAD (Jiang et al., 2023) and GenAD (Zheng et al., 2025) do not perform tracking, predicting trajectories based on the temporal information embedded within the BEV feature. The main issue with these methods is that they attempt to use a single feature (query) to represent an object's appearance and motion. Compared to pure semantic learning, motion occupies a portion of the feature channels but fails to contribute to perception, resulting a negative transfer in the perception module. Our work effectively addresses this issue.

## 3    Method

Figure 2 shows an overview of DMAD structure. Sensor embeddings are extracted from multi-view camera images and are shared across all tasks, including detection, tracking, mapping, prediction, and planning. We initialize three distinct types of queries—object, map, and motion—which attend to the sensor embeddings to extract the specific information required for each respective task. Based on the type of information learned, the decoding process is divided into two pathways. On one way, object and map decoding are jointly performed within the **Interactive semantic decoder**, where both types of queries iteratively exchange latent semantic information at each decoding layer. On the other way, motion queries extract motion information from the sensor embeddings within the **Neural-Bayes motion decoder**. Each motion query

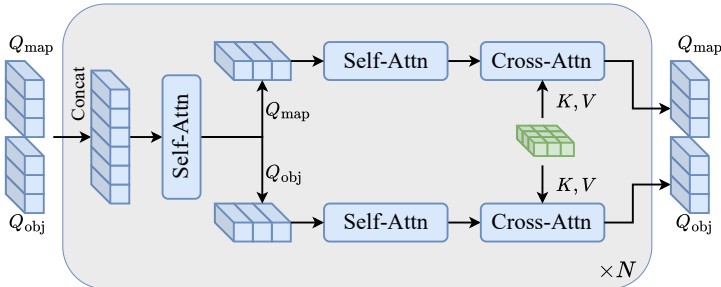

Figure 3: **Interactive semantic decoding.** Object and map queries are concatenated and interact through a self-attention module before being separated to independently attend to the sensor embeddings. This process is repeated across $N$ stacked layers.

is paired with an object query, using the object's coordinates as a reference point at each decoding layer. After decoding each frame, the motion query's predicted future waypoint becomes the object query's reference point in the next frame, similar to the recursion of a Bayes filter (Thrun et al., 2005). The exchange of reference points is always without gradient. At last, the motion queries are passed on to the planning module. The system is fully E2E trainable, with motion and semantic gradients propagated in distinct paths.

## 3.1 Interactive Semantic Decoder

To leverage the semantic correlation between individual objects and map elements, we introduce the Interactive Semantic Decoder. In contrast to the unidirectional interaction in DualAD (Doll et al., 2024), our approach enables a bidirectional exchange of information.

We initialize a set of object queries $Q_{\text{obj}} = \{q_{\text{obj},n} \in \mathbb{R}^d\}_{n=0}^{N_{\text{obj}}-1}$ and a set of map queries $Q_{\text{map}} = \{q_{\text{map},n} \in \mathbb{R}^d\}_{n=0}^{N_{\text{map}}-1}$. The number of queries could be different, while the dimensions $d$ must be the same. Each decoding layer first concatenates both types of queries. Self-attention (Vaswani et al., 2017) is then applied, where both tasks exchange their semantic information. Subsequently, the two types of queries are divided, each performing self-attention and cross-attention on the sensor embeddings, respectively, as shown in Fig. 3.

After interactive semantic decoding, each object query is classified into a category $c$ and regressed into a vector $[\Delta x, \Delta y, \Delta z, w, h, l, \theta]^\mathsf{T}$, where $(\Delta x, \Delta y, \Delta z)$ represents the coordinate offset to the query's reference point $[x_{\text{ref}}, y_{\text{ref}}, z_{\text{ref}}]^\mathsf{T}$, $w, h, l$ are the width, height, and length of the object, and $\theta$ indicates the heading. Rather than directly learning the absolute coordinates of the object, it learns the offsets relative to its corresponding reference points. Thus, the bounding boxes can be represented as $[x_{\text{ref}} + \Delta x, y_{\text{ref}} + \Delta y, z_{\text{ref}} + \Delta z, w, h, l, \theta]^\mathsf{T}$. Notably, velocities are not regressed, as they pertain to motion information. We design the object queries to focus solely on semantic information, *i.e.*, the object's category, center point, size, and orientation.

## 3.2 Neural-Bayes Motion Decoder

We introduce a novel motion decoder operating in parallel with the semantic decoder, aimed at fully decoupling motion and semantic learning to reduce the negative transfer in semantic tasks. Given the correlation between motion and semantics, we design a recursive process to facilitate the exchange of human-readable information between the two decoders as illustrated in Fig. 4, which comprises the processes of prediction, measurement, and updating, similar to the Bayes filter (Thrun et al., 2005). Appendix C provides a brief introduction to the Bayes filter. We proceed with the elaboration of the proposed motion decoder.

**Initialization.** We initialize a set of motion queries $Q_{\text{mt}} = \{q_{\text{mt},n} \in \mathbb{R}^d\}_{n=0}^{N_{\text{mt}}-1}$ in the same way we initialize object queries. The motion queries correspond one-to-one with the object queries, *i.e.*, $N_{\text{mt}} = N_{\text{obj}}$. However, since they do not directly interact in the latent space, their dimensionalities $d$ can differ. Each motion query represents the motion state of an object, although the model does not initially know whether

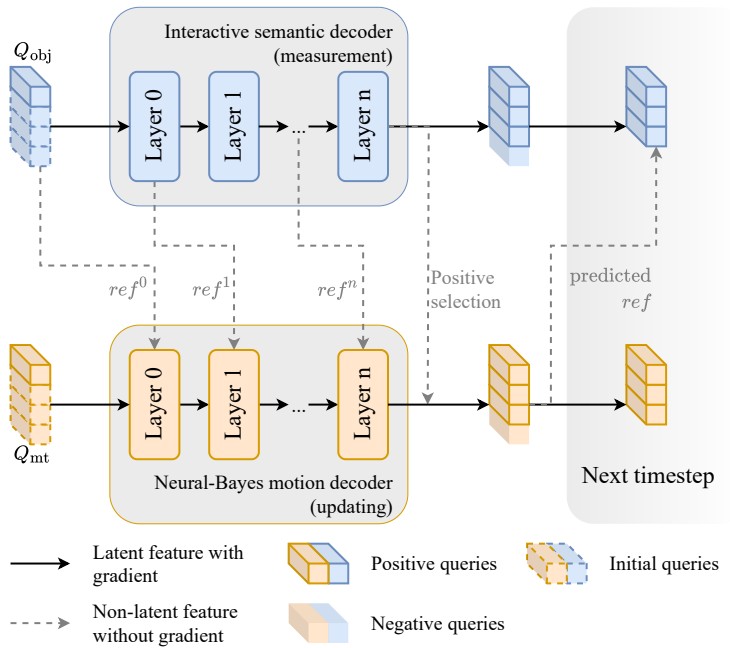

Figure 4: **Neural-Bayes motion decoding.** After each decoding layer, the semantic decoder updates the reference points, which are then shared with the motion decoder. At the end of each frame, positive object query indices are used to select corresponding motion queries and are together propagated to the subsequent frame, with the motion query predictions serving as reference points for the next frame. This process is similar to the measurement, updating, and prediction steps in a Bayes filter. Map queries, ego queries and sensor embeddings are omitted for simplicity.

the object exists. Additionally, motion queries and object queries share a common set of reference points.

**Measurement.** The detection, already introduced in Sec. 3.1, is treated as the measurement in Bayes filter. After each semantic decoding layer, the object queries are regressed, yielding the coordinate vectors $ref = [x, y, z]^\mathsf{T}$ of the tentative object, which then serves as reference points for the next layer:

$$ref^{l+1} = f_{\text{reg}}(f^l_{\text{Semantic-Dec}}(Q^l_{\text{obj}}, Z, ref^l)), \tag{1}$$

where the superscript denotes the layer and $Z$ is the sensor embeddings.

**Updating.** With the reference points $ref^l$ from the semantic decoding (the inter-layer reference points update in Fig. 2), the motion queries also attend to the sensor embeddings via cross-attention:

$$Q^{l+1}_{\text{mt}} = f^l_{\text{Motion-Dec}}(Q^l_{\text{mt}}, Z, ref^l), \tag{2}$$

where the motion queries are updated conditioned on the measured reference points.

**Prediction.** The prediction occurs in two stages: first through the unimodal trajectory construction, followed by the multimodal prediction.

The first stage estimates the unimodal trajectory via an MLP:

$$T_{\text{uni}} = \{s_t\}^{t_{\text{fut-uni}}}_{t=t_{past}} = f_{\text{MLP-uni}}(q_{\text{mt}}), \tag{3}$$

where $q_{\text{mt}} \in \mathbb{R}^d$ is a single motion query, and $s_t$ represents the waypoint $[x, y]^\mathsf{T}$ at timestep $t$. It produces a single trajectory that spans from the past timestep $t_{\text{past}}$ to the future timestep $t_{\text{fut-uni}}$.

We calculate the velocity at $t = 0$ via finite difference:

$$v_0 = \frac{s_1 - s_{-1}}{2\Delta t},\tag{4}$$

where $\Delta t$ indicates the time interval of a timestep.

We use $s_1$ as the reference point for the subsequent timestep, *i.e.*, inter-frame reference points update in Fig. 2, for object tracking.

In the second stage, the motion query attends to the sensor embeddings and is then decoded into $K$ future trajectories over the next $t_{\text{fut-multi}}$ timesteps, along with their corresponding confidence scores:

$$(\{T_k\}_{k=1}^K, \{c_k\}_{k=1}^K) = f_{MLP}(\text{Cross-Attn}(q_{\text{mt}}, Z)),\tag{5}$$

where $c_k$ represents the confidence score of the $k$-th trajectory.

**Tracking.** Multi-object tracking is performed using the query propagation mechanism (Zeng et al., 2022; Lin et al., 2023). Each object query is associated with a unique instance ID. A positive query propagates across consecutive frames, ensuring that corresponding detections are assigned the same ID. During training, object queries associated with ground truth are referred to as positive queries; during inference, positivity is determined by whether the confidence score exceeds a specified threshold:

$$Q_{t+1} = \begin{cases} \{F_{propagate}(q_t^{id}) \mid q_t^{id} \in Q_t \wedge \text{IsMatched}(q_t^{id}, \text{GT})\}, & \text{in training} \\ \\ \{F_{propagate}(q_t^{id}) \mid q_t^{id} \in Q_t \wedge c_t^{id} > \tau\}, & \text{in inference} \end{cases},\tag{6}$$

where GT represents ground-truth objects and $\tau$ stands for the positive threshold. The propagation of motion queries follows that of object queries, as they are related. Queries with the same $id$ correspond to the same object instance, whose decoded bounding boxes or trajectories represent the state of this instance across different timesteps. This propagation mechanism enables continuous measuring, updating, and predicting, similar to a Bayes filter.

## 4 Experiments

We conduct experiments on the nuScenes (Caesar et al., 2020) dataset to validate the effectiveness of our method. We present results in three parts. The first part focuses on perception (detection, tracking, and mapping). In the second part, we evaluate motion prediction and planning. Lastly, we provide an extensive ablation study and SHAP values (Lundberg & Lee, 2017) visualization.

### 4.1 Training Configuration

We reproduce UniAD (Hu et al., 2023) and SparseDrive (Sun et al., 2024) as baselines. To demonstrate the effectiveness and generality of our approach, we integrate it into both frameworks, resulting in two variants named DMAD and SparseDMAD. Both baselines utilize the query propagation mechanism; however, UniAD extracts dense BEV features from image inputs, while SparseDrive employs sparse scene representations. Besides the aforementioned tasks, UniAD additionally performs occupancy prediction. We also retain the occupancy module in comparisons with UniAD for task consistency. As occupancy prediction serves merely as another representation of upstream tasks, we describe it in Appendix D. We adhere as closely as possible to default configurations of the baseline; however, to ensure a rigorous comparison, some adjustments are made. The following paragraphs outline the adjustments and the rationale behind them.

| Method | NDS↑ | mAP↑ | mAVE↓ |
|---|---|---|---|
| VAD (Jiang et al., 2023) | 0.460 | 0.330 | 0.405 |
| GenAD (Zheng et al., 2025) | 0.280 | 0.213 | 0.669 |
| PARA-Drive (Weng et al., 2024) | 0.480 | 0.370 | - |
| UniAD - stage 1 | 0.497 | 0.382 | 0.411 |
| UniAD - stage 2 | 0.491 (-1.2%) | 0.377 (-1.3%) | 0.412 (+0.2%) |
| DMAD - stage 1 | 0.504 | 0.395 | 0.406 |
| DMAD - stage 2 | 0.506 (+0.4%) | 0.396 (+0.3%) | 0.395 (-2.7%) |
| SparseDrive - stage 1 | 0.531 | 0.419 | 0.257 |
| SparseDrive - stage 2 | 0.523 (-1.5%) | 0.417 (-0.5%) | 0.269 (+4.7%) |
| SparseDMAD - stage 1 | **0.536** | 0.424 | 0.260 |
| SparseDMAD - stage 2 | 0.534 (-0.4%) | **0.427** (+0.7%) | **0.253** (-2.7%) |

Table 1: **Object detection results.** The performance changes in stage 2 are expressed as percentages, with red indicating a decline and blue representing improvement.

| Method | AMOTA↑ | AMOTP↓ | IDS↓ |
|---|---|---|---|
| ViP3D (Gu et al., 2023) | 0.217 | 1.63 | - |
| MUTR3D (Zhang et al., 2022) | 0.294 | 1.50 | 3822 |
| PARA-Drive (Weng et al., 2024) | 0.350 | - | - |
| UniAD - stage 1 | 0.374 | 1.31 | 816 |
| UniAD - stage 2 | 0.354 (-5.3%) | 1.34 (+2.3%) | 1381 (+69%) |
| DMAD - stage 1 | 0.394 | 1.32 | 781 |
| DMAD - stage 2 | 0.393 (-0.3%) | 1.30 (-1.5%) | 767 (-1.8%) |
| SparseDrive - stage 1 | 0.395 | 1.25 | 602 |
| SparseDrive - stage 2 | 0.376 (-4.8%) | 1.26 (+0.8%) | **559** (-7.1%) |
| SparseDMAD - stage 1 | **0.396** | **1.23** | 608 |
| SparseDMAD - stage 2 | 0.395 (-0.3%) | **1.23** (0%) | 571 (-6.1%) |

Table 2: **Multi-object tracking results.**

**Two-stage training.** We follow the two-stage training scheme of our baseline. In the first stage, we train object detection, tracking, and mapping. In the second stage, we train all modules together. Notably, because our tracking relies on reference points provided by unimodal prediction, we incorporate unimodal prediction training in the first stage. Multimodal prediction is trained only in the second stage, which is consistent with the baseline.

**Queue length.** Since AD is a time-dependent task, the model typically processes a sequence of consecutive frames as a training sample. The number of input frames, *i.e.*, the queue length $q$, defines the temporal horizon the model can capture, impacting the performance of related tasks. UniAD employs different queue lengths across its two training stages: 5 in the first stage and 3 in the second. The reduced queue length in the second stage degrades perception performance due to reduced temporal aggregation, shown in Appendix F. This degradation hinders the identification of negative transfer effects caused by the sequential structure. To mitigate this interference, we standardize the queue length to 3 across both training stages in comparisons with UniAD. Unless otherwise specified, the performance of UniAD in all result tables is reproduced with a queue length of 3 using the official codebase (UniAD-contributors, 2023). SparseDrive does not have this issue, and we use the default setting of 4.

| Method | Lanes↑ | Drivable↑ | Dividers↑ |
|---|---|---|---|
| BEVFormer (Li et al., 2022) | 0.239 | **0.775** | - |
| PARA-Drive (Weng et al., 2024) | **0.330** | 0.710 | - |
| UniAD - stage 1 | 0.293 | 0.650 | 0.248 |
| UniAD - stage 2 | 0.312 (+6.5%) | 0.678 (+4.3%) | 0.267 (+7.7%) |
| DMAD - stage 1 | 0.292 | 0.655 | 0.242 |
| DMAD - stage 2 | 0.321 (+9.9%) | 0.691 (+5.5%) | **0.271** (+12%) |

(a) Map segmentation results.

| Method | $AP_{ped}$↑ | $AP_{divider}$↑ | $AP_{boundary}$↑ | mAP↑ |
|---|---|---|---|---|
| MapTR (Liao et al., 2023) | **0.562** | 0.598 | 0.601 | **0.587** |
| VAD (Jiang et al., 2023) | 0.406 | 0.515 | 0.506 | 0.476 |
| SparseDrive - stage 1 | 0.533 | 0.579 | 0.575 | 0.562 |
| SparseDrive - stage 2 | 0.494 (-7.3%) | 0.569 (-1.7%) | 0.583 (+1.4%) | 0.549 (-2.3%) |
| SparseDMAD - stage 1 | 0.553 | 0.599 | **0.606** | 0.586 |
| SparseDMAD - stage 2 | 0.554 (+0.2%) | **0.601** (+0.3%) | **0.606** (0%) | **0.587** (+0.2%) |

(b) Vectorized mapping results.

Table 3: **Map perception results.**

| Method | EPA↑ | | minADE↓ | |
|---|---|---|---|---|
| | C | P | C | P |
| ViP3D (Gu et al., 2023) | 0.226 | - | 2.05 | - |
| GenAD (Zheng et al., 2025) | **0.588** | 0.352 | 0.84 | 0.84 |
| UniAD | 0.495 | 0.361 | 0.69 | 0.79 |
| DMAD | 0.535 | **0.416** | 0.72 | 0.77 |
| SparseDrive | 0.487 | 0.406 | **0.63** | 0.73 |
| SparseDMAD | 0.500 | 0.410 | **0.63** | **0.71** |

Table 4: **Trajectory prediction results.** C and P stand for cars and pedestrians respectively.

**Ego query** represents the features directly used for motion planning, which is intended to capture the motion information of the ego vehicle. SparseDrive generates the ego query from the front camera image and the estimated previous ego status, which blends semantics and motion, thus contradicting our dividing design. To align with our proposal, we eliminate the use of the front image for the ego query when applying DMAD to SparseDrive. For UniAD, we retain the planning module unchanged, as it initializes the ego query randomly.

## 4.2 Perception

**Metrics.** For object detection and tracking, we use the metrics defined in the nuScenes benchmark. The primary metrics for detection are nuScenes Detection Score (NDS) and mean average precision (mAP). For multiple object tracking, we report the average multi-object tracking accuracy (AMOTA) and the average multi-object tracking precision (AMOTP). For map segmentation, we use the intersection over union (IoU) metric of drivable areas, lanes, and dividers. Vectorized mapping adopts mAP of lane divider, pedestrian crossing and road boundary.

**Object detection.** Table 1 presents the detection performance across two training stages. In the first stage, thanks to the interactive semantic decoding, our approach slightly outperforms the baseline. After the second stage of training, baseline's performance shows a decline. In contrast, our method preserves the

| Method | Perception tasks | Ego states in planner | $L_2$ distances (m) ↓ | | | | Collision rates (%) ↓ | | | |
|---|---|---|---|---|---|---|---|---|---|---|
| | | | 1s | 2s | 3s | Avg. | 1s | 2s | 3s | Avg. |
| Ego-MLP (Zhai et al., 2023) | ✗ | ✓ | 0.17 | 0.34 | 0.60 | 0.370 | $0^\dagger$ | $0.27^\dagger$ | $0.85^\dagger$ | $0.373^\dagger$ |
| AD-MLP (Li et al., 2024) | ✗ | ✓ | 0.14 | 0.10 | 0.41 | 0.217 | 0.10 | 0.10 | 0.17 | 0.123 |
| VAD (Jiang et al., 2023) | ✓ | ✗ | 0.41 | 0.70 | 1.05 | 0.720 | 0.07 | 0.17 | 0.41 | 0.217 |
| DualVAD (Doll et al., 2024) | ✓ | ✗ | 0.30 | 0.53 | 0.82 | 0.550 | 0.11 | 0.19 | 0.36 | 0.220 |
| GenAD (Zheng et al., 2025) | ✓ | ✗ | 0.28 | 0.49 | 0.78 | 0.517 | 0.08 | 0.14 | 0.34 | 0.187 |
| UniAD* (Hu et al., 2023) | ✓ | ✗ | 0.42 | 0.63 | 0.91 | 0.656 | 0.07 | 0.10 | 0.22 | 0.130 |
| PARA-Drive (Weng et al., 2024) | ✓ | ✗ | **0.25** | **0.46** | **0.74** | **0.483** | 0.14 | 0.23 | 0.39 | 0.253 |
| UniAD | ✓ | ✗ | 0.48 | 0.76 | 1.12 | 0.784 | 0.07 | 0.11 | 0.27 | 0.150 |
| DMAD | ✓ | ✗ | 0.38 | 0.60 | 0.89 | 0.625 | 0.07 | 0.12 | **0.19** | 0.127 |
| SparseDrive | ✓ | ✗ | 0.32 | 0.61 | 1.00 | 0.643 | 0.01 | **0.06** | 0.22 | 0.097 |
| SparseDMAD | ✓ | ✗ | 0.30 | 0.61 | 1.01 | 0.643 | **0** | 0.07 | 0.21 | **0.093** |

Table 5: **Open-loop planning.** Ego-MLP and AD-MLP are faded since both learn only the ego motion. *Results from the checkpoint in the official repository (UniAD-contributors, 2023), trained with a queue length of 5 in stage 1. †Ego-MLP employs a different strategy in the evaluation of collision rates, therefore the results are not comparable. We reproduce SparseDrive using the official code, but the results differ from its paper because some errors have been fixed after publication.

| Method | NeuroNCAP scores ↑ | | | | Collision rates (%) ↓ | | | |
|---|---|---|---|---|---|---|---|---|
| | Stat. | Frontal | Side | Avg. | Stat. | Frontal | Side | Avg. |
| UniAD | 3.50 | 1.17 | 1.67 | 2.11 | 32.4 | 77.6 | 71.2 | 60.4 |
| DMAD | 4.40 | 1.47 | 2.07 | 2.65 | **14.8** | 74.0 | 61.6 | 50.1 |
| SparseDrive | 4.42 | 2.96 | 2.30 | 3.23 | 22.4 | 62.8 | 60.4 | 48.5 |
| SparseDMAD | **4.57** | **3.14** | **2.42** | **3.37** | 18.4 | **60.0** | **59.1** | **45.8** |

Table 6: **Closed-loop planning.** We use the official implementation of NeuroNCAP, but our results differ from those in the original paper because the codebase has been updated since its publication.

perceptual performance of the first stage, benefiting from separated motion learning that mitigates negative transfer. Ultimately, our method surpasses UniAD and SparseDrive by 3.1% and 2.1% in NDS, respectively.

**Multi-object tracking.** Due to using a single feature vector to represent semantics and motion, UniAD and SparseDrive exhibit negative transfer of 5.3% and 4.8% in AMOTA, as shown in Tab. 2. Our dividing design enables object queries to learn about appearance more effectively. At the same time, unimodal predictions offer enhanced tracking reference points. Consequently, our method achieves a gain of 11.0% and 5.1% in AMOTA, respectively.

**Map perception.** UniAD does not encounter negative transfer in map segmentation. Leveraging the advantages of interactive semantic decoding, our method marginally surpasses UniAD. Our method mitigates the negative transfer in vectorized online mapping, significantly surpassing SparseDrive by 7.0% in mAP, (see Tab. 3).

### 4.3 Prediction and Planning

**Metrics.** For prediction, we use E2E perception accuracy (EPA) proposed in ViP3D (Gu et al., 2023) as the main metric. We also report the minimum average displacement error (minADE). However, since minADE can only be computed for true positive detections, it does not fully capture the predictive capabilities of the E2E system, whereas EPA accounts for the number of false positives. For open-loop planning, we use $L_2$ distances and collision rates. Moreover, we evaluate driving safety in a closed-loop environment using NeuroNCAP (Ljungbergh et al., 2024). This framework reconstructs scenes from the nuScenes dataset and

| Method ID | Interactive semantic dec. | Motion queries | Inter-layer ref. update | Inter-frame ref. update | #Params (M) | Inference time (s) | NDS↑ | AMOTA↑ | Lanes↑ | EPA↑ | Avg. $L_2$↓ | Avg. Col.↓ |
|---|---|---|---|---|---|---|---|---|---|---|---|---|
| 1 (UniAD) | ✗ | ✗ | ✗ | ✗ | 127.3 | 0.47 | 0.491 | 0.354 | 0.312 | 0.495 | 0.784 | 0.150 |
| 2 | ✓ | ✗ | ✗ | ✗ | 128.0 | 0.48 | 0.503 | 0.382 | 0.320 | 0.524 | 0.683 | 0.150 |
| 3 | ✗ | ✓ | ✓ | ✓ | 139.3 | 0.49 | 0.502 | 0.387 | 0.313 | **0.535** | 0.661 | 0.143 |
| 4 | ✓ | ✓ | ✗ | ✗ | 140.4 | 0.49 | 0.481 | 0.339 | 0.322 | 0.485 | 0.655 | 0.163 |
| 5 | ✓ | ✓ | ✓ | ✗ | 140.4 | 0.49 | 0.489 | 0.352 | **0.323** | 0.498 | 0.648 | 0.160 |
| 6 | ✓ | ✓ | ✗ | ✓ | 140.4 | 0.49 | 0.495 | 0.364 | 0.319 | 0.512 | 0.631 | 0.137 |
| 7 (DMAD) | ✓ | ✓ | ✓ | ✓ | 140.4 | 0.49 | **0.506** | **0.393** | 0.321 | **0.535** | **0.625** | **0.127** |

Table 7: **Ablation of DMAD.**

inserts safety-critical objects. The resulting scores are derived from collision rates and impact speeds.

**Trajectory prediction.** We report car and pedestrian prediction metrics in Tab. 4. Our method surpasses both baselines in EPA, especially achieving improvements of 0.040 for cars and 0.055 for pedestrians over UniAD. However, our method does not improve the minADE of cars. One possible reason is that once detection performance exceeds a certain threshold, further detection improvements often come from reducing false negatives of challenging objects that are either distant or occluded. These hard-to-detect objects typically have limited historical motion data and larger coordinate errors, making them more difficult to predict. A similar issue is observed in UniAD (Hu et al., 2023): in the supplementary materials, UniAD-Large substantially surpasses UniAD-Base in EPA (thanks to better detection and tracking performance), yet it falls short of UniAD-Base in minADE.

**Planning.** For open-loop evaluation, we adopt the evaluation method of VAD (Jiang et al., 2023), which accommodates the widest range of models to our knowledge. We report our results in Tab. 5. Notably, jointly optimizing $L_2$ distances and collision rates proves challenging. While PARA-Drive achieves the lowest $L_2$ distances, it also exhibits the highest collision rates. In the closed-loop evaluation, our structure benefits both baselines in all three cases with stationary, frontal, and side critical objects. We validate that the improvements in perception can be propagated to planning, achieving SOTA collision rates and NeuroNCAP Scores.

## 4.4 Ablation Study

We ablate our proposed decoders, as shown in Tab. 7, decomposing the motion decoder into three components: motion query, inter-layer, and inter-frame reference point updating.

**Model profile.** In methods with multi-view camera images as inputs, the primary computational cost is concentrated in the image backbone (Li et al., 2022). In contrast, our approach focuses on the decoding component, resulting in minimal impact on model size and inference speed. Compared to UniAD (Hu et al., 2023), our decoders add 13.1M parameters and increase inference latency by 0.02 seconds on an NVIDIA RTX 6000 Ada.

**Effect of dividing and merging.** Experiments ID 1, 2, 3, 7 demonstrate the effectiveness of both proposed decoders. The standalone application of the interactive semantic decoder (ID 2) significantly enhances the performance of object detection, tracking, and map segmentation. The standalone application of the Neural-Bayes motion decoder (ID 3) markedly improves prediction and planning. Notably, ID 3 also significantly enhances detection and tracking, attributed to freeing object queries from learning velocities and the higher-quality reference points provided by the unimodal prediction. Experiments ID 4, 5, 6, 7 show the importance of inter-layer and inter-frame updating in the Neural-Bayes motion decoder.

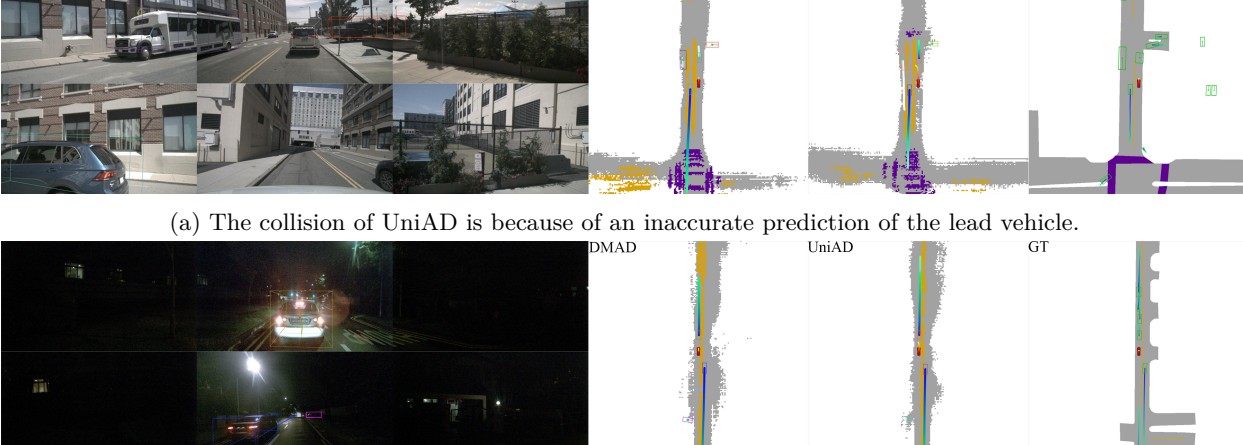

(a) The collision of UniAD is because of an inaccurate prediction of the lead vehicle.

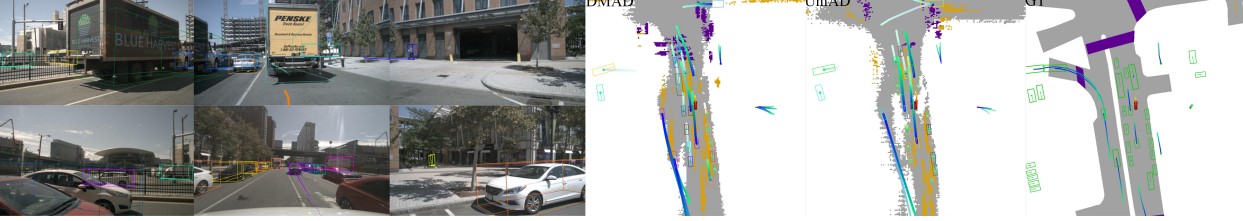

(b) Both models make inaccurate predictions of the lead vehicle during the night. However, UniAD collides with the lead vehicle due to its aggressive driving policy.

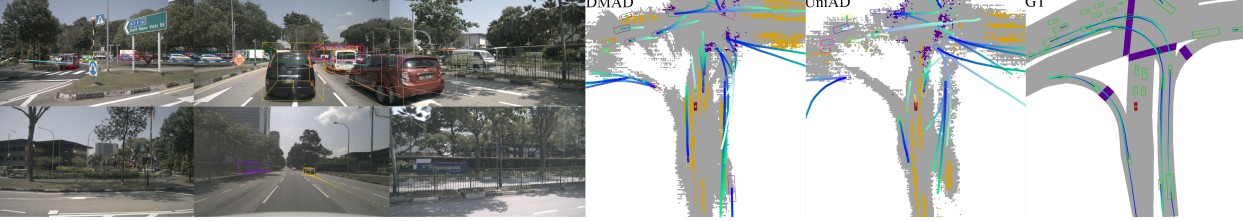

(c) An inaccurate detection (the detected position is too close to the ego vehicle) causes yielding, and then colliding with another vehicle.

(d) UniAD fails to detect the lead vehicle and collides with it.

Figure 5: **Qualitative comparison between DMAD and UniAD.** Each subfigure demonstrates a sample where UniAD encounters collision while DMAD does not.

## 4.5 Visualizations

We provide qualitative comparisons between DMAD and UniAD in Fig. 5, showcasing how the improved perception and prediction reduces collision rates. More visualizations can be found in Appendix J.

We use SHAP values (Lundberg & Lee, 2017)—which quantify the contribution of each feature to the change in a model's output—to inspect the negative transfer in detection and tracking. We visualize the SHAP values of the object query with respect to the object classification output. Changes in SHAP values across the two training stages reveal the negative transfer in UniAD and highlight the effectiveness of our method.

Figure 6a compares the SHAP values between stage 1 and stage 2 of UniAD, sorted in descending order. The left half of the difference bar chart predominantly shows negative values, whereas the right half shows positive values. This indicates that SHAP values in stage 1 are more uniformly distributed, while those in stage 2 are more concentrated. Compared with a flat distribution, this concentration indicates that fewer features are contributing to the classification task, reducing detection and tracking performance. This observation aligns

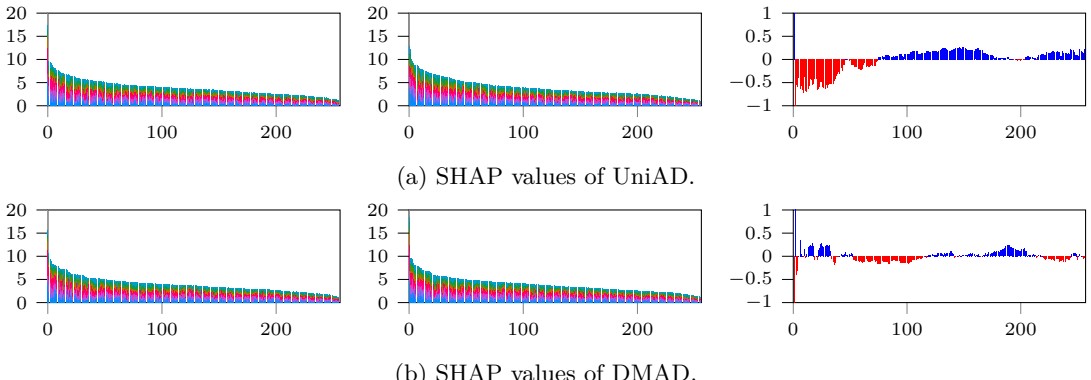

(a) SHAP values of UniAD.

(b) SHAP values of DMAD.

Figure 6: **SHAP values of stage 1 (left), stage 2 (middle), and the difference (right).** Each bar represents the SHAP values of a single feature with respect to different classes. The object query consists of 256 features, forming 256 bars in each chart. The difference is computed as stage 1 minus stage 2, aggregating all classes, where red indicates a negative value and blue signifies a positive value.

with our argument that during the second stage, object queries are expected to learn motion information, which does not benefit the perception task. Specifically, while the velocity learned in stage 1 is sufficient for tracking (predicting the next timestep), it is inadequate for the long-term prediction over 12 timesteps (6 seconds). Therefore, the object query is forced to learn more motion states that offer limited utility for identifying objects, interfering with the space for semantic information. In contrast, the SHAP values in DMAD maintain a similar distribution across both stages, as shown in Fig. 6b.

## 5 Conclusion

In this work, we show that by decoupling semantic and motion learning, we eliminate the negative transfer that E2E training typically imposes on object and map perception. Besides, we leverage the correlation between semantic tasks to promote positive transfer during E2E training. We validate that our improvements in perception and prediction directly enhance planning performance, achieving SOTA collision rates. However, our method is fundamentally built upon the query propagation mechanism, where the motion query is propagated across timesteps and continuously updated to represent the motion feature of an object. In contrast, architectures like VAD (Jiang et al., 2023) freshly initialize all queries at every timestep, making it impossible to continuously monitor the motion of an object. This lack of query propagation makes our approach incompatible without a major redesign. Addressing this limitation remains future work.

**Acknowledgments:** The authors thank Stellantis, Opel Automobile GmbH for the fruitful collaboration and the opportunity to contribute to the "STADT:up" project (FKZ 19A22006P) funded partly by the German Federal Ministry for Economic Affairs and Energy (BMWE) and the European Union. The authors also thank the project "NXT GEN AI METHODS" funded by the BMWK and gratefully acknowledge the computing resources supported by the Helmholtz Association's Initiative and Networking Fund on HAICORE@FZJ and by the Federal Ministry of Education and Research and the Ministry of Science, Research and Arts Baden-Württemberg on HoreKa at NHR@KIT.

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

# A Intermediate Tasks vs. Planning Performance

The final objective of our method is to enhance E2E planning performance, which we consider the most crucial metric for evaluating an AD system. Our focus on intermediate tasks, such as detection and tracking, is therefore not an end in itself but an approach to achieve this goal. We posit that by improving the performance of these intermediate stages, we can more effectively guide the optimization of the downstream planning module.

This perspective is consistent with the evolution of the field. While early end-to-end models (Pomerleau, 1988) omit intermediate outputs, recent E2E methods incorporate them as auxiliary tasks (Chitta et al., 2021; Zeng et al., 2019). This approach offers two key advantages: first, it provides interpretability for the E2E system, which is valuable for verification and safety; second, it guides the optimization process of the model toward a better local optimum, yielding superior driving performance.

Within this context, our work distinguishes itself by its different focus. Instead of asking what new auxiliary tasks can be added to improve planning, we investigate how to optimally utilize existing ones. By improving the learning process of these intermediate modules, we demonstrate an effective pathway to enhancing the overall performance of E2E AD systems.

# B Tracking as a Semantic Task

We justify the similarity of detection and tracking on nuScenes (Caesar et al., 2020) by analyzing the information learned by the object query. E2E detection and tracking models decode each query into category, location, size, orientation, and velocity. The category is clearly a semantic attribute, while location, size, and orientation serve as spatial complements to the category, all being time-invariant. In contrast, velocity is derived from time, making it a motion attribute. However, measuring velocities is not a common practice in detection, but required by the nuScenes benchmark. Therefore, detection models trained on nuScenes are able to perform tracking without any additional learning effort assuming constant velocity motion (Zhang et al., 2022; Hu et al., 2023; Lin et al., 2023; Gu et al., 2023). Given that current modular E2E models are all trained on nuScenes, we regard the tracking in these methods closely resembles detection, where learning semantics is dominating.

# C Bayes Filter

Bayes filter (Thrun et al., 2005) estimates an unknown distribution based on the process model and noisy measurements as follows:

$$p(\mathbf{x}_t \mid \mathbf{z}_{1:t}) \propto p(\mathbf{z}_t \mid \mathbf{x}_t)\, p(\mathbf{x}_t \mid \mathbf{z}_{1:t-1}), \tag{7}$$

where $\mathbf{x}$ denotes the state, $\mathbf{z}$ represents the measurement, and the subscript indicates timesteps. The task is to estimate the state $\mathbf{x}_t$ at timestep $t$ given all the measurements $\mathbf{z}_{1:t}$ in the past from timestep 1 to $t$, which is proportional to the product of the likelihood $p(\mathbf{z}_t \mid \mathbf{x}_t)$ and the prediction $p(\mathbf{x}_t \mid \mathbf{z}_{1:t-1})$.

Some well-known instances of Bayes filter, *e.g.*, Kalman filter, are widely used in traditional object tracking. The tracking process can be carried out in three steps: first, predicting the current position based on the object's historical states $\mathbf{x}_{1:t-1}$; second, identifying the observation that best matches the prediction (data association); finally, updating the current state $\mathbf{x}_t$ according to the latest measurement $\mathbf{z}_t$. This process is recursively executed over successive timesteps. We find semantics and motion correspond to the measurement and state in Bayes filter, respectively. Therefore, we introduce the architecture of Bayes filter to transformer decoders, resulting in Neural-Bayes motion decoder.

# D Occupancy Prediction

We retain the occupancy prediction module from UniAD to ensure task consistency, where the BEV feature serves as the query and learns from motion prediction features (output queries) through cross-attention.

Consequently, we regard occupancy prediction in UniAD as a secondary task to perception and motion prediction, as it merely offers an alternative representation of upstream tasks.

DMAD achieves similar performance ($\text{IoU}_{\text{near}}$: 62.7%, $\text{IoU}_{\text{far}}$: 39.8%) to UniAD ($\text{IoU}_{\text{near}}$: 62.9%, $\text{IoU}_{\text{far}}$: 39.6%). The advances of DMAD in upstream tasks do not generalize to occupancy prediction. The reason could be that, by dividing semantics and motion, output features of the prediction module lack spatial information desired by occupancy prediction, such as size, whereas output features of UniAD's prediction module preserve the spatial information.

## E Implementation Details

The implementations of our proposed interactive semantic decoder and neural-bayes motion decoder vary slightly when applied to different baselines. This is intentional, as our goal is to preserve the original architecture of each baseline as much as possible. Consequently, we adapt our decoders by making minor modifications to their hyperparameters to fit each specific baseline, rather than altering the baselines to unify the implementation of our decoders. Therefore, we introduce DMAD (our implementation based on UniAD) and SparseDMAD (our implementation based on SparseDrive) separately.

### E.1 DMAD

**Interactive semantic decoder.** The interactive semantic decoder is implemented by inserting 6 transformer layers before the standard detection decoders of the baseline models. Each layer is composed of a multi-head self-attention module and a feed-forward network (FFN), with layer normalization applied after each module.

The multi-head self-attention module is configured with 8 heads, an embedding dimension of 256, and a dropout rate of 0.1. The FFN consists of two linear layers with an intermediate ReLU activation, which expands the dimension from 256 to an inner-layer dimension of 512 before projecting it back to 256.

During the forward pass, the object and map queries are concatenated and processed sequentially through these 6 transformer layers. The output is then split back into updated object and map queries, which are subsequently fed into the standard decoders for the respective tasks of object detection and map perception.

**Neural-Bayes motion decoder.** The motion decoder shares the same architecture as the detection decoder, comprising 6 sequential layers. Each layer performs the following sequence of operations: self-attention, layer normalization, cross-attention, layer normalization, a FFN, and another layer normalization.

The self-attention module is a standard multi-head attention mechanism configured with 8 heads, a dropout rate of 0.1, and an embedding dimension of 256.

The cross-attention is implemented using deformable attention, which allows each query to adaptively aggregate spatial features from the BEV representation. It utilizes two distinct linear layers to estimate sampling offsets with respect to a reference point and the corresponding attention weights. For each query, features are sampled from 4 points and combined via a weighted sum. This module is also configured with 8 heads and a dropout rate of 0.1.

The FFN consists of two linear layers with an intermediate ReLU activation. It expands the feature dimension from 256 to a hidden dimension of 512 and then projects it back to 256. A dropout rate of 0.1 is also applied.

Finally, the unimodal trajectory prediction is performed by a 2-layer MLP with ReLU activation. Although predictions are generated and supervised by the ground-truth at each decoder layer, only the output from the final layer is used for updating reference points.

### E.2 SparseDMAD

**Interactive semantic decoder.** The SparseDrive baseline employs non-homogeneous decoding layers: one beginning layer without temporal feature aggregation, followed by five layers that do incorporate it. Based

| Queue length stage 1 | Queue length stage 2 | NDS↑ | mAP↑ | AMOTA↑ | AMOTP↓ | IDS↓ | Lanes↑ | Drivable↑ | EPA↑ | minADE↓ | Avg. $L_2$↓ | Avg. Col.↓ |
|---|---|---|---|---|---|---|---|---|---|---|---|---|
| 3 | 3 | 0.491 | 0.377 | 0.354 | 1.34 | 1381 | 0.312 | 0.678 | **0.495** | 0.692 | 0.784 | 0.150 |
| 5 | 3 | 0.499 | 0.381 | 0.362 | 1.34 | 956 | 0.313 | **0.692** | 0.492 | **0.655** | 0.656 | 0.130 |
| 5 | 5 | **0.501** | **0.384** | **0.370** | **1.32** | **885** | **0.314** | 0.690 | **0.495** | 0.714 | **0.615** | **0.123** |

Table 8: **Effect of queue length on UniAD.**

on empirical results, we insert our interaction module only into these latter 5 layers, as this configuration yields superior performance.

The original operational sequence in these five layers is: [temporal-cross-attn, self-attn, norm, deformable-attn, norm, ffn, norm]. Our modification involves inserting our interaction module before the deformable attention module. The interaction module consists of a self-attention, a layer normalization, a FFN, and another layer normalization. This implementation strategy differs from that of DMAD but is chosen to align with the architectural style of SparseDrive.

The self-attention module in our inserted block is configured with 8 heads, an embedding dimension of 512, and a dropout rate of 0.1. The embedding dimension is twice the query dimension because, in SparseDrive and SparseDMAD, the positional embedding is concatenated with the query embedding, rather than added as in UniAD and DMAD.

The FFN consists of two linear layers with an intermediate ReLU activation, a hidden dimension of 1024, and a dropout rate of 0.1.

**Neural-Bayes motion decoder.** In SparseDMAD, the motion decoder also duplicates the architecture of the detection decoder, consisting of 6 non-homogeneous layers. The first layer operates without temporal cross-attention, while the subsequent five layers perform the full operational sequence: [temporal-cross-attn, self-attn, norm, deformable-attn, norm, ffn, norm].

The attention modules within these layers serve distinct purposes. Temporal cross-attention allows current motion queries to attend to historical ones to aggregate temporal information. Self-attention models the interactions among the current set of motion queries. Finally, deformable attention enables the motion queries to sample features from the BEV embeddings. All three attention modules are configured with 8 heads, an embedding dimension of 512, and a dropout rate of 0.1.

The FFNs are composed of two linear layers with an intermediate ReLU activation, a hidden dimension of 1024, and a dropout rate of 0.1. All normalization layers are standard layer normalizations.

The motion queries are decoded by a 3-layer MLP. The input and intermediate embedding dimension is 256 and it uses ReLU as the activation function.

## F  Queue Length

We adopt a different queue length configuration from that of the original UniAD. As mentioned in Sec. 4.1, the rationale behind our decision is that reducing the queue length in stage 2 affects the performance, hindering the observation of negative transfer. Table 8 shows an ablation study of queue length on UniAD, presenting the performance drops by reduced queue length. As the training time scales almost linearly to the queue length, we opt for a queue length of 3 to reduce training time of each iteration.

## G  Effect of Unimodal Prediction Horizon

We conduct experiments on the number of future steps in unimodal prediction, as shown in Tab. 9. We observe that the unimodal prediction horizon influences the proportion of motion information within the BEV feature, thereby impacting the performance of both semantic and motion tasks. A long prediction horizon degrades the performance of semantic tasks, as the BEV feature is forced to prioritize motion learning in order to predict distant future outcomes. Experiments show that a prediction horizon of 6 seconds minimizes

| Unimodal pred. horizon | NDS↑ | mAP↑ | AMOTA↑ | AMOTP↓ | IDS↓ | Lanes↑ | Drivable↑ | EPA↑ | minADE↓ | Avg. $L_2$↓ | Avg. Col.↓ |
|---|---|---|---|---|---|---|---|---|---|---|---|
| 2s | **0.516** | **0.404** | **0.400** | **1.30** | **695** | 0.321 | 0.691 | 0.534 | 0.735 | 0.679 | 0.220 |
| 4s | 0.506 | 0.396 | 0.393 | **1.30** | 767 | 0.321 | 0.691 | **0.535** | **0.723** | **0.625** | 0.127 |
| 6s | 0.504 | 0.396 | 0.384 | **1.30** | 751 | **0.322** | **0.700** | 0.525 | 0.743 | 0.629 | **0.117** |

Table 9: **Effect of unimodal prediction horizon on DMAD.**

| Obj-mt | Mt-map | Obj-map | NDS↑ | mAP↑ | AMOTA↑ | AMOTP↓ | IDS↓ | Lanes↑ | Drivable↑ | EPA↑ | minADE↓ | Avg. $L_2$↓ | Avg. Col.↓ |
|---|---|---|---|---|---|---|---|---|---|---|---|---|---|
| ✓ | ✗ | ✗ | 0.490 | 0.375 | 0.360 | 1.31 | 981 | 0.311 | 0.682 | 0.498 | 0.740 | 0.763 | 0.160 |
| ✗ | ✓ | ✗ | 0.501 | 0.383 | 0.381 | 1.31 | 801 | 0.320 | **0.691** | 0.511 | **0.719** | 0.676 | 0.147 |
| ✗ | ✗ | ✓ | **0.506** | **0.396** | **0.393** | **1.30** | **767** | 0.321 | **0.691** | **0.535** | 0.723 | **0.625** | **0.127** |

Table 10: **Effect of query interactions.**

the collision rates, but performs worst in tracking. Although this phenomenon can also be referred to as negative transfer, our approach is unable to address this specific type, as the BEV feature is shared across all tasks and is expected to encapsulate both types of information. To balance motion and semantic information within the BEV feature, we set the prediction horizon to 4 seconds.

## H  Effect of Query Interactions

In addition to the primary object-map interaction used in DMAD, we investigate two alternative designs: object-motion and motion-map interaction. We find that both of these alternative designs re-establish a direct gradient flow between the motion and semantic tasks. This conflicts with our "divide" concept and results in suboptimal performance, as detailed in Tab. 10. Specifically, enabling interaction between object and motion queries degrades the model's overall performance to the baseline level. The interaction between motion and map queries, while also detrimental, introduces a less severe degradation.

## I  Effect of Velocity Estimation

We investigate several alternative designs for velocity estimation in DMAD, comparing four methods: (1) regressing velocity from object queries (the standard baseline method), (2) deriving velocity from bounding box positions, (3) regressing from motion queries, and (4) deriving it from predicted trajectories. The results are presented in Tab. 11.

While the choice of velocity estimation method has a minor impact on overall perception performance, this comparison offers valuable insights. The analysis reveals two key findings: first, removing the burden of motion learning from the object query improves performance on perception tasks. Second, it demonstrates the benefit of using the motion query for this task.

Methods (3) and (4) exhibit similar performance. However, we select method (4) because it does not require an additional MLP head to decode the motion query, making it a more efficient choice.

## J  Visualizations and Failure Cases

**Interaction visualizations between object and map queries.** We use the attention heatmap to visualize the interaction between object and map queries in the interactive semantic decoder in Fig. 7. We notice different behaviors in different layers. In Layer 0, most queries are newly initialized (except for the propagated object queries from the previous frame), so the heatmap seems like random noise. In the subsequent layers, we can see how information exchange happens between object and map queries. The object queries attend to map queries strongly in Layers 1 and 5. The map queries attend to object queries strongly in all layers from 1 to 5. As the layer depth increases, the focus shifts more towards individual object queries. It is worth noting that object queries with an index above 900 (the area immediately to the

| Velocity estimation method | mAVE↓ | NDS↑ | mAP↑ | AMOTA↑ | AMOTP↓ | IDS↓ | Lanes↑ | Drivable↑ |
|---|---|---|---|---|---|---|---|---|
| Regressed from $Q_{\mathrm{obj}}$ | 0.411 | 0.502 | 0.393 | 0.389 | **1.30** | 812 | 0.320 | **0.691** |
| Derived from bounding box positions | 0.541 | 0.506 | 0.395 | 0.390 | **1.30** | 798 | 0.318 | 0.689 |
| Regressed from $Q_{\mathrm{mt}}$ | **0.395** | **0.507** | **0.396** | 0.392 | **1.30** | **752** | 0.320 | 0.690 |
| Derived from unimodal trajectories | **0.395** | 0.506 | **0.396** | **0.393** | **1.30** | 767 | **0.321** | **0.691** |

Table 11: **Effect of velocity estimation.**

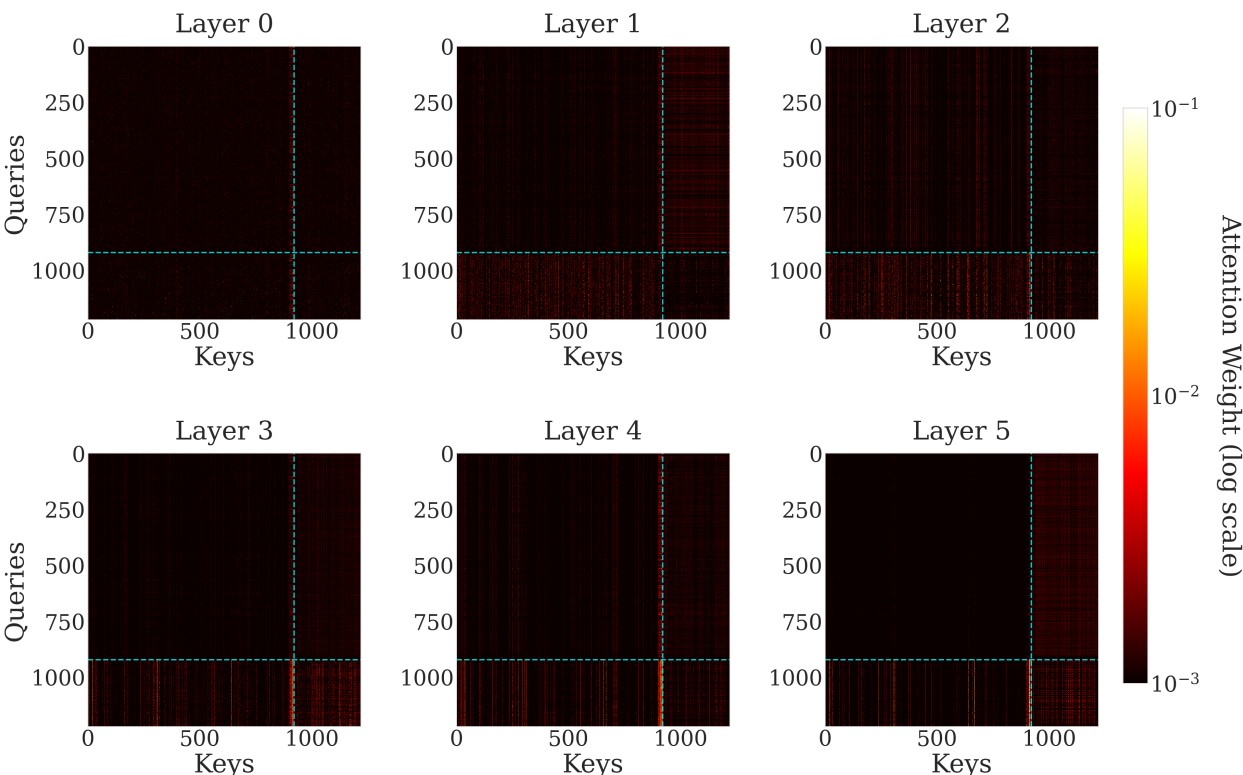

Figure 7: **Interaction visualization between object and map queries.** We plot the attention heatmaps of the 6 Self-Attn layers right after the concatenation of object and map queries. We use cyan dashed lines to divide a heatmap into four regions: the **upper-left** shows the attention between object queries, the **upper-right** shows the attention from object queries to map queries, the **lower-left** shows the attention from map queries to object queries, and the **lower-right** shows the attention between map queries.

left of the vertical cyan dashed lines) receive significantly more attention from map queries. We hypothesize that this is because these queries are propagated from the previous frame and possessed high confidence scores; consequently, map queries are more inclined to gather information from these high-confidence object queries.

**Model output visualizations and failure cases.** We provide more visualizations of DMAD during the day (Figs. 8 to 10), in rainy weather (Figs. 11 and 12), at night (Figs. 13 to 15), and for failure cases (Figs. 16 to 19).

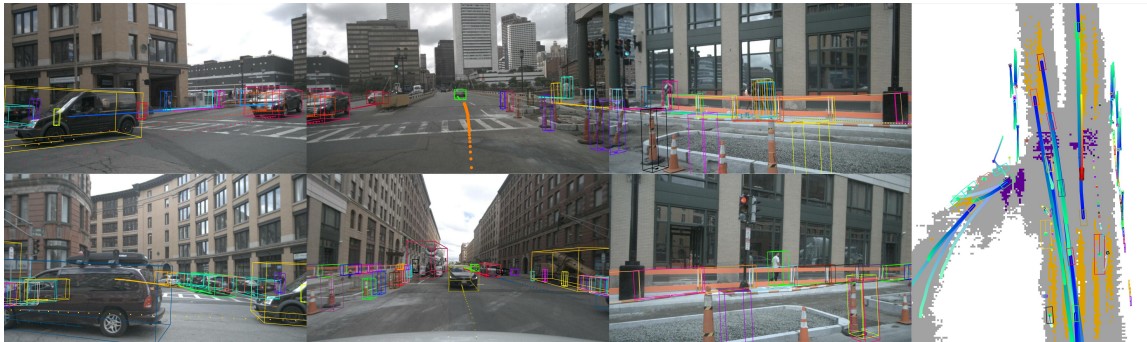

Figure 8: **Visualization of DMAD.** Cloudy, going straight

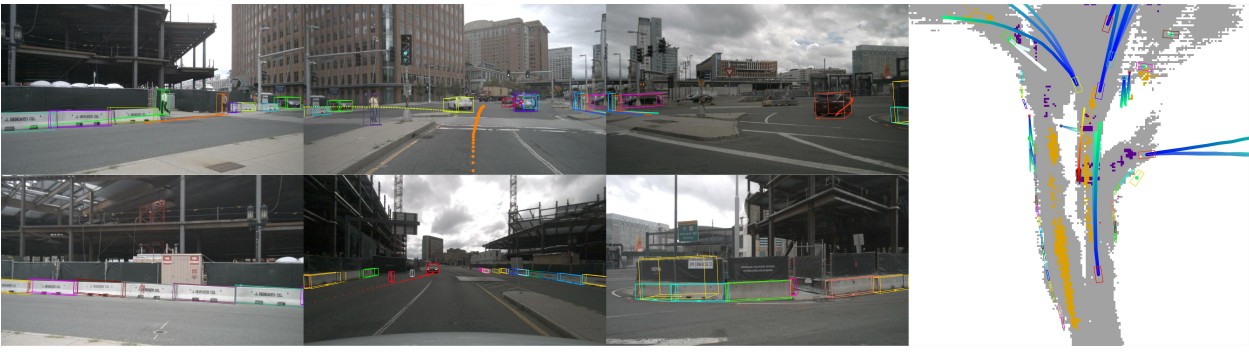

Figure 9: **Visualization of DMAD.** Cloudy, going straight.

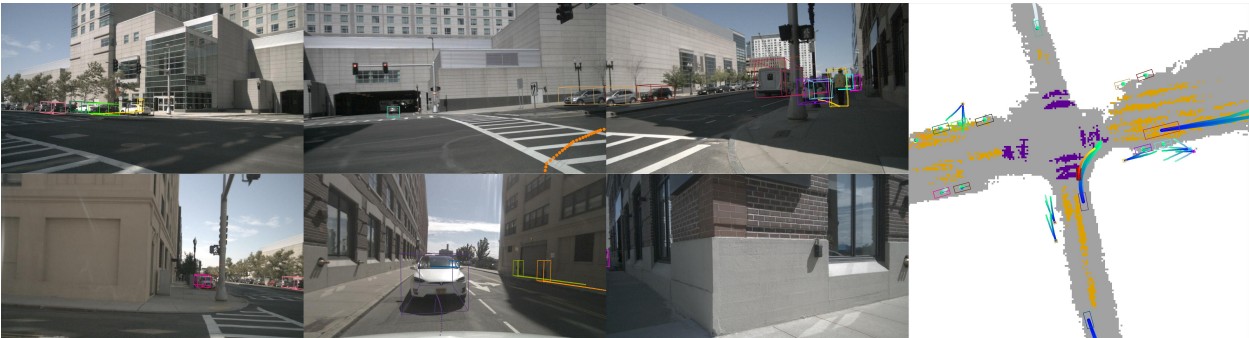

Figure 10: **Visualization of DMAD.** Sunny, turning right.

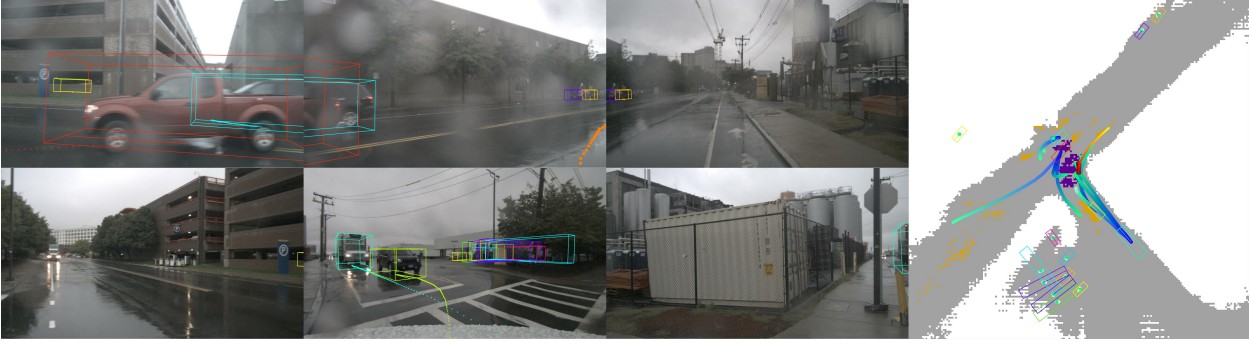

Figure 11: **Visualization of DMAD.** Rainy, turning right.

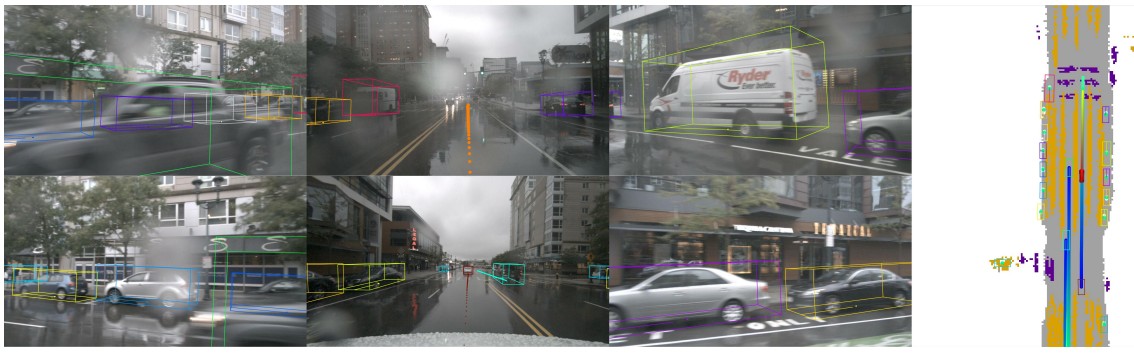

Figure 12: **Visualization of DMAD.** Rainy, turning right.

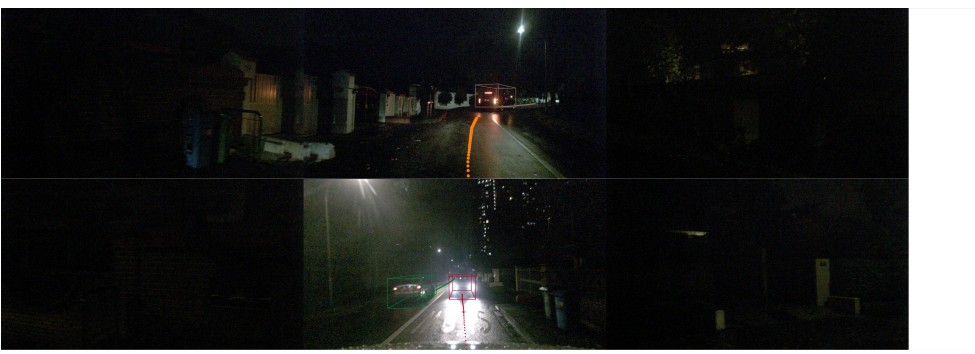

Figure 13: **Visualization of DMAD.** Night, going straight.

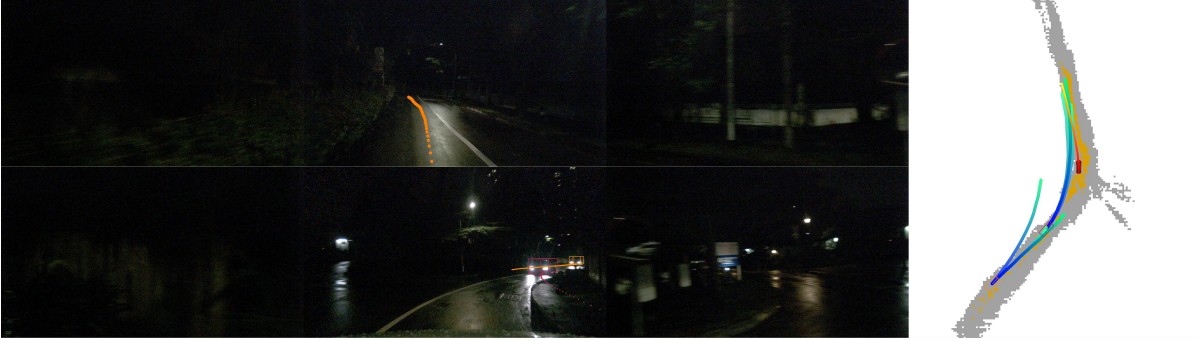

Figure 14: **Visualization of DMAD.** Night, going straight.

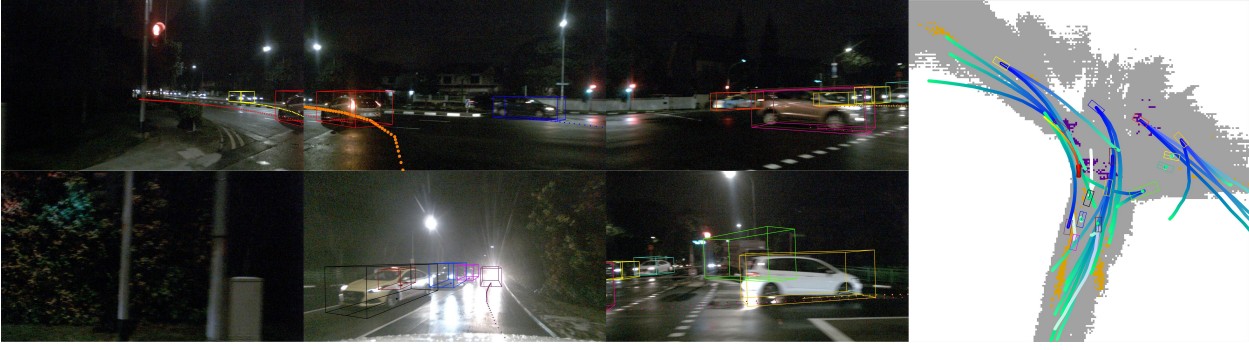

Figure 15: **Visualization of DMAD.** Night, turning left.

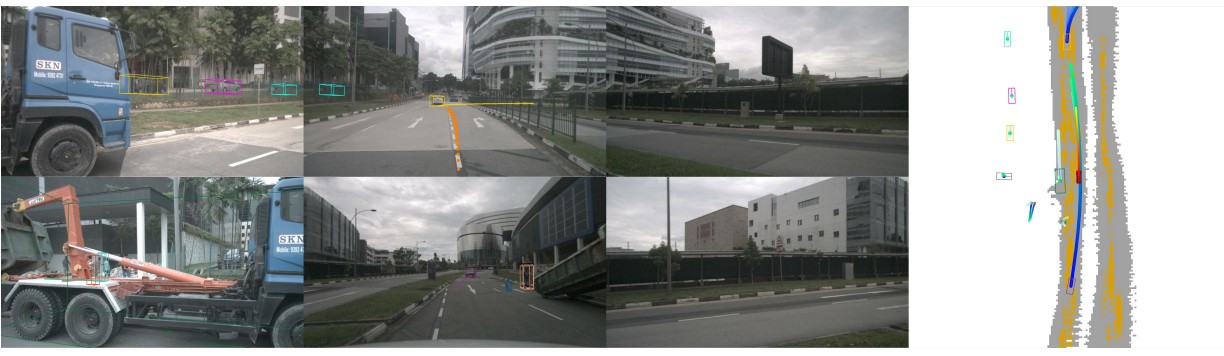

Figure 16: **Failure case of DMAD.** Driving between two lanes.

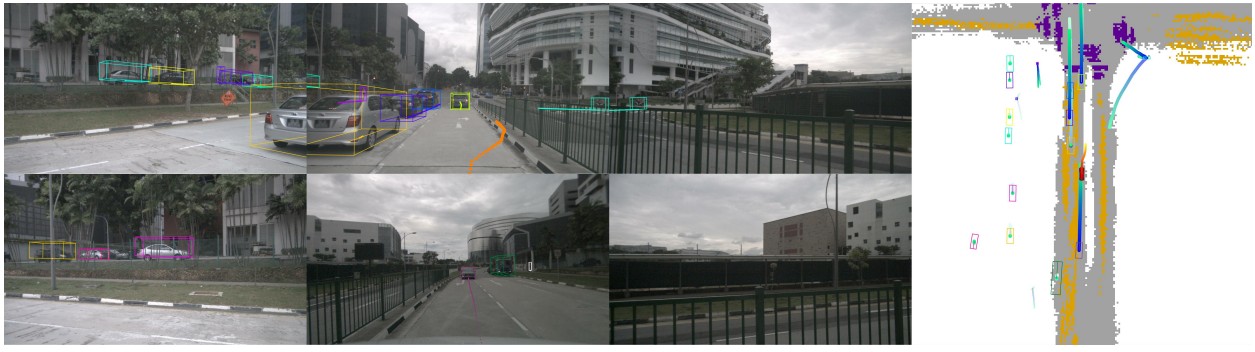

Figure 17: **Failure case of DMAD.** Going out of drivable area.

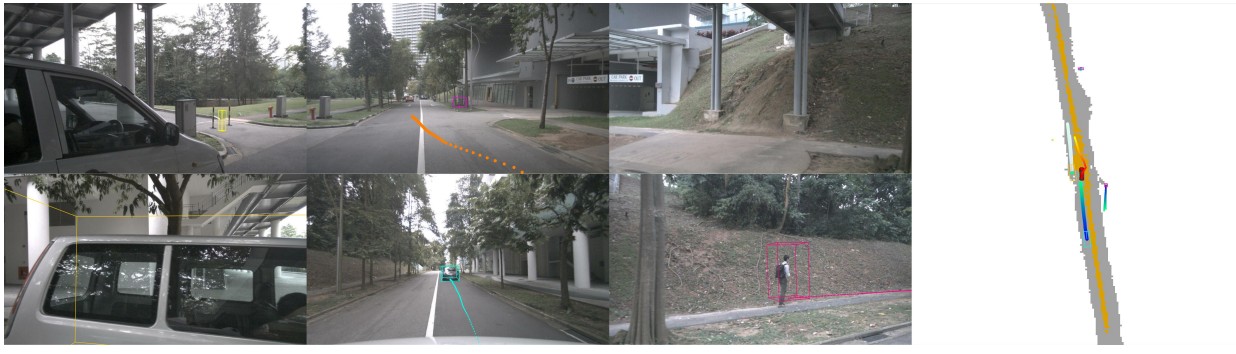

Figure 18: **Failure case of DMAD.** Giving too much space when overtaking.

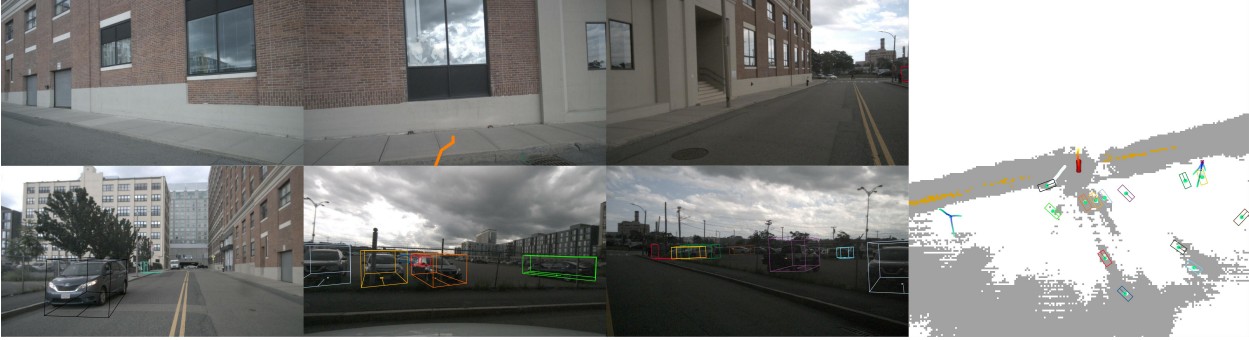

Figure 19: **Failure case of DMAD.** Unable to reverse or turn around.

