# OpenReview forum: "Divide and Merge: Motion and Semantic Learning in End-to-End Autonomous Driving"
_TMLR — Accepted by TMLR_

### Review · Reviewer_MLfT · 2025-08-13

**Summary Of Contributions:**

This work describes an autonomous driving system with components for detection, tracking, motion prediction and planning.  Motion prediction and planning are mostly separated from detection and tracking (as well as map estimates) heads, with the only interaction between them being an interchange of updated point estimates, which is done in a pipelined way through the depth of transformer heads in addition to between frames.  In this way, improved estimates can improve performance of each part of system, but without struggling with performance degradations of individual components that arise from jointly training competing objectives in fully end-to-end systems.  Evaluations are performed showing competitive results with baseline systems UniAD, SparseDrive and others on a range of benchmarks.  Notably, lower-level detect and track metrics still improve when training with all objectives in stage 2.

**Additional Comments:**

Some other questions I had while reading the paper:

- sec 3.1: how are object and map queries initialized?  and how are the reference points obtained?

- sec 3.2:  how are motion queries initialized?

- what is the advantage of using both concat with self-attention mixing, as well as subsequent alternating self- and cross-attention mixing?  including both of these styles, as opposed to twice as many of one or the other, seems unnecessary but it's possible there could in practice be an advantage

**Audience:**

Yes

**Audience Explanation:**

The paper presents arguments for how to make use of interactions between det+track and motion prediction, develops an architecture accordingly, and shows behavior (particularly non-degradation or improvement of all metrics when trained jointly) that is in-line with its design.

**Broader Impact Concerns:**

A discussion of the impact within the larger context of AD may be warranted here.  In particular, since there is a focus on keeping intermediate det+track metrics up, what is the impact of these in the larger AD context, versus focusing entirely on end planning metrics?

**Claims And Evidence:**

Yes

**Claims Explanation:**

The system is evaluated well on several benchmarks with ablations that support the claims, with a couple caveats that I think could be cleared up before acceptance:

1.  Experiments showing stage 2 training improves stage 1 for det+track is not compared to training stage 1 objectives for longer, so it's unclear if the higher performance for these objectives is due to benefits from improved stage 2 motion reference point estimates, or simply training stage 1 objectives for longer, or a combination of both.  However, it clearly shows there is not a degradation --- so the claim of no degradation is already well supported, just not yet the source of improvement.

2.  I didn't see an evaluation that tried to reenable backpropagation between motion - detection interactions, e.g. just by reenabling backprop through the exchanged reference points e.g. in a position encoded format, though this may also have difficulties in implementation for this architecture

**Requested Changes:**

See (1) in Claims section above:  In order to claim improvement from joint training, I think this needs to compare to training longer without motion prediction/planning objectives.  However, the claim of non-degradation from adding motion (which is somewhat weaker than mutually reinforcing objectives) has already been demonstrated well.

---

> ### Author Response · Authors · 2025-09-04
>
> Thank you for your feedback on our work. Below are our responses to the concerns you raised.
>
> > Q1: In order to claim improvement from joint training, I think this needs to compare to training longer without motion prediction/planning objectives. However, the claim of non-degradation from adding motion (which is somewhat weaker than mutually reinforcing objectives) has already been demonstrated well.
>
> We would like to clarify that our manuscript does not claim mutual reinforcement between _perception and motion tasks_, but rather between _object perception and online mapping_. This mutual reinforcement is already established during the stage 1 training, where our method demonstrates superior perception and mapping performance compared to the baselines. This result supports our argument for mutual reinforcement.
>
> > Q2: I didn't see an evaluation that tried to reenable backpropagation between motion - detection interactions, e.g. just by reenabling backprop through the exchanged reference points e.g. in a position encoded format, though this may also have difficulties in implementation for this architecture
>
> We briefly explored this direction but again encountered negative transfer in perception; therefore, we did not go further.
>
> Specifically, we implemented an interaction between object and motion queries via self-attention, similar to the interaction between object and map queries in our proposed interactive semantic decoder. As a result, the detection and tracking performance fell back to a level comparable to that of UniAD.
>
> > Q3: A discussion of the impact within the larger context of AD may be warranted here. In particular, since there is a focus on keeping intermediate det+track metrics up, what is the impact of these in the larger AD context, versus focusing entirely on end planning metrics?
>
> That is a good question. In short, our focus is not purely on the intermediate metrics themselves. Instead, we aim to indirectly improve the final planning performance by enhancing the performance of intermediate tasks. To this end, we evaluated our method on both open-loop and closed-loop planning benchmarks comprehensively.
>
> We believe that for the task of autonomous driving, planning performance is the most important. While initial E2E research proposed solutions without intermediate results [1], incorporating intermediate or auxiliary tasks has become a mainstream practice in recent years [2,3]. This is because, in E2E systems, intermediate results on one hand provide a degree of interpretability, and on the other hand, they act as auxiliary tasks to guide the planning module's optimization toward a better local optimum, thereby achieving better driving performance.
>
> However, most of these methods have been limited to exploring _what_ auxiliary tasks can be added to improve planning. Our work, in contrast, focuses on _how_ to optimally utilize these auxiliary tasks.
>
> > Q4: sec 3.1: how are object and map queries initialized? and how are the reference points obtained?
>
> We follow the initialization method of the baselines: learnable queries and positional embeddings are initialized using `torch.nn.Embedding()`, with weights sampled from a standard normal distribution. A single linear layer is then used to convert positional embeddings into reference points.
>
> > Q5: sec 3.2: how are motion queries initialized?
>
> Same as object and map queries.
>
> > Q6: what is the advantage of using both concat with self-attention mixing, as well as subsequent alternating self- and cross-attention mixing? including both of these styles, as opposed to twice as many of one or the other, seems unnecessary but it's possible there could in practice be an advantage
>
> The advantage is that this implementation is simple yet effective.
>
> The semantic decoding layer consists of three key steps. First, a concat with self-attention mixing facilitates information exchange between the object and map queries. Second, a standard self-attention module processes interactions within each query type separately (i.e., object-to-object and map-to-map). Third, a cross-attention module extracts relevant features from the image.
>
> The latter two steps are standard practice in detection transformers. Our contribution, based on the concept of merging, is the addition of the first step. The concat with self-attention mixing is a simple implementation of this concept, yet it has proven effective.
>
> [1] Pomerleau, Dean A. "Alvinn: An autonomous land vehicle in a neural network." _Advances in neural information processing systems_. 1988.
>
> [2] Zeng, Wenyuan, et al. "End-to-end interpretable neural motion planner." _Proceedings of the IEEE/CVF conference on computer vision and pattern recognition_. 2019.
>
> [3] Chitta, Kashyap, et al. "Neat: Neural attention fields for end-to-end autonomous driving." _Proceedings of the IEEE/CVF International Conference on Computer Vision_. 2021.

---

> > ### Comment · Reviewer_MLfT · 2025-09-10
> >
> > Thanks for the clarifications, I think these address most of my concerns.  For Q1, while I now see in the intro an a few other scattered sentences that just object+map are related, the first bullet point at the end of sec 1 doesn't mention these tasks by name and is a little unclear; it could be good to touch this up.  For Q2, the fact that you saw negative transfer actually supports your argument for separation, in my opinion, and could be good to put some of these results in an ablation or appendix.

---

> > > ### Author Response · Authors · 2025-09-11
> > >
> > > Thank you for the suggestion. We have revised the manuscript accordingly and uploaded the revised version.

---

### Review · Reviewer_pEyw · 2025-08-14

**Summary Of Contributions:**

This paper proposes a solution for the negative transfer problem encountered when the E2E autonomous driving (AD) framework is jointly trained on detection, tracking, motion prediction, and planning tasks. The solution includes two modules.
The first module, the interactive semantic decoder, concatenates the object and map queries and applies self-attention layers between them, allowing the bidirectional information exchange between the objects and map elements.
The second module, Neural-Bayes motion decoder, passes the updated object locations into the motion decoder across different layers, and uses the motion prediction as the initialization of object queries at the next timestep. This design resembles a Bayes filter for detecting and tracking objects.
These two modules are implemented over UniAD and SparseDrive, respectively. The experiments show that they have brought improvements on multiple tasks across the AD scenarios. Experiments also show that the proposed changes reduce the negative transfer in the 2-stage training procedure.

**Audience:**

Yes

**Audience Explanation:**

* The proposed changes are simple, interesting, and seem to carry novel insights. Overall, the proposed changes allow and encourage the exchange of information in different decoding modules for different tasks. The motion decoder passes the information between the semantic decoder and the motion decoder, forming a novel Bayes filtering framework for detection, tracking, and motion prediction.

**Claims And Evidence:**

Yes

**Claims Explanation:**

* The proposed method is integrated into and extensively tested with two existing E2E AD methods, UniAD and SparseDrive. The experiments show that the proposed changes can enhance the performance on almost all AD perception and planning tasks over the baselines.
* The proposed method only imposes a minimal overhead in terms of model parameters and computation latency.
* The ablation study verifies the effectiveness of each proposed design. It's interesting to see that the interactive semantic decoder alone can bring a large improvement across the board.
* An analysis of the SHAP value is provided, which provides a possible explanation of negative transfer.

**Requested Changes:**

* The writing in the introduction and background sections needs to be improved. I think the authors need to use some text to more clearly describe the design of the recent E2E AD framework. Especially, I think the design of shared embedding and task-specific query can be more clearly introduced. Also, the negative transfer problem and the SHAP value analysis can be put forward, which gives a better motivation and background.
* Despite the performance gains, the motivation and the mechanism of the proposed changes are unclear. Why do the object and map queries need to go through a self-attention layer? Are there alternative designs (e.g., performing self-attention over more queries from more tasks)? Why could this self-attention layer result in such a performance gain? Why is the measurement step done per-layer, but the updating step is done per-timestep? Are there alternative designs for the Bayes decoder (e.g., a self-attention layer between object and motion queries)? Also, exactly what kind of information is exchanged in both decoders? Do the reference points used in the updating step actually make sense? I think more analyses, visualizations, and studies on alternative designs can help extract more insights from the proposed changes.

I think this paper presents an interesting method, thorough experiments, and strong results.

As mentioned above, I would suggest changes to improve writing and adding more analytical experiments that can reveal the mechanism and insights of the proposed method.

---

> ### Author Response · Authors · 2025-09-04
>
> Thank you for your feedback on our work. Below are our responses to the concerns you raised.
>
> > Q1.1: I think the authors need to use some text to more clearly describe the design of the recent E2E AD framework. Especially, I think the design of shared embedding and task-specific query can be more clearly introduced.
>
> Thanks for pointing out the missing description of shared embeddings. Actually we described it in the abstract but somehow missed it in the introduction. We will add it in the first paragraph of introduction.
>
> > Q1.2: Also, the negative transfer problem and the SHAP value analysis can be put forward, which gives a better motivation and background.
>
> We are not sure if we get your point since in the introduction chapter we are already straight to the point of negative transfer problem. The information flow of the introcution is:
> - background: modular E2E AD is multi-task learning
> - problem: the negative transfer in multi-task learning
> - analysis: sequentially learning semantics and motion (in a shared embedding) is the cause of negative transfer
> - proposed solution: the divide and merge structure
>
> We appreciate it if you could clarify your sugguestion based on the obove information flow.
>
> > Q2.1: Why do the object and map queries need to go through a self-attention layer?
>
> This self-attention layer enables information exchange between object and map queries. It is a simple implementation of the concept of "merge".
>
> > Q2.2: Are there alternative designs (e.g., performing self-attention over more queries from more tasks)?
>
> Yes, but they are suboptimal. We explored object-map, object-motion, and motion-map self-attention. The object-map brings the bigest benefits to the overall performance.
>
> > Q2.3: Why could this self-attention layer result in such a performance gain?
>
> The performance gain is because the added self-attention layer explicitly models the synergistic relationship between objects and the map. For instance, a "lane" context helps to identify "vehicles", while the presence of "vehicles" helps to identify a "drivable area".
>
> > Q2.4: Why is the measurement step done per-layer, but the updating step is done per-timestep?
>
> Both measurement and updating are done per-layer. This is known as deep supervision, which is standard practice for detection transformers.
>
> The prediction is done per-timestep.
>
> > Q2.5: Are there alternative designs for the Bayes decoder (e.g., a self-attention layer between object and motion queries)?
>
> See the response to Q2.2.
>
> > Q2.6: Also, exactly what kind of information is exchanged in both decoders?
>
> The exchanged information is: the reference point (x,y,z) of each query, and the index of positive detections. It's shown in Figure 4.
>
> > Q2.7: Do the reference points used in the updating step actually make sense?
>
> Yes. These reference points are supervised by the ground truth of the object's position.
>
> > Q2.8: I think more analyses, visualizations, and studies on alternative designs can help extract more insights from the proposed changes.
>
> Thanks for the sugguestion. We will add the followings in the appendix:
> - query interaction analysis as mentioned in Q2.2.
> - altenative designs in velocity estimation: (1) derive from bounding box positions (2) regression based on object queries (3) regression based on motion queries (4) derived from predicted trajectories. This set of experiments reveals the positive effects on perception tasks after removing the burden of velocity (motion) learning from the object query, as well as the benefits of using the motion query for velocity estimation.
> - more visualizations, especially failure cases

---

> > ### Comment · Reviewer_pEyw · 2025-09-30
> >
> > Thank the authors for the comprehensive response! Most of my concerns are addressed and here are some minor remaining ones.
> >
> > > We appreciate it if you could clarify your sugguestion based on the obove information flow.
> >
> > I mostly meant section 4.5 can be put before or at the beginning of the method section.
> >
> > > Yes, but they are suboptimal. We explored object-map, object-motion, and motion-map self-attention. The object-map brings the bigest benefits to the overall performance.
> >
> > Do you have quantitative or qualitative results somewhere?
> >
> > > Both measurement and updating are done per-layer. This is known as deep supervision, which is standard practice for detection transformers.
> >
> > But according to Figure 4, "**After each decoding layer**, the semantic decoder updates the reference points, which are then shared with the motion decoder. **At the end of each frame**, positive object query indices are used to select corresponding motion queries and are together propagated to the subsequent frame"
> >
> > > This self-attention layer enables information exchange between object and map queries. It is a simple implementation of the concept of "merge".
> > > The performance gain is because the added self-attention layer explicitly models the synergistic relationship between objects and the map. For instance, a "lane" context helps to identify "vehicles", while the presence of "vehicles" helps to identify a "drivable area".
> >
> > I didn't find evidence for these two claims. While they are quite possible according to the network design, evidence like visualization, case study, or ablation experiments are needed to reveal what the network is really doing. I encourage the authors to provide such evidence in the next revision.

---

> > > ### Author Response · Authors · 2025-10-14
> > >
> > > Thank you for your feedback. Although the official discussion period has concluded and a decision has been reached, we would like to briefly clarify these points for the record.
> > >
> > > > I mostly meant section 4.5 can be put before or at the beginning of the method section.
> > >
> > > Thanks for the suggestion. We considered this ordering carefully, but we believe it would be better to keep the current order since we need to introduce the method before showing the SHAP values of the method.
> > >
> > > > Do you have quantitative or qualitative results somewhere?
> > >
> > > Yes, they have been added in Table 10.
> > >
> > > > But according to Figure 4, "After each decoding layer, the semantic decoder updates the reference points, which are then shared with the motion decoder. At the end of each frame, positive object query indices are used to select corresponding motion queries and are together propagated to the subsequent frame"
> > >
> > > We'd clarify that the measurement and updating is layer-wise. The prediction/propagating is frame-wise:
> > >
> > > "**After each decoding layer**, the semantic decoder **updates** the reference points, which are then shared with the motion decoder. **At the end of each frame**, positive object query indices are used to select corresponding motion queries and are together **propagated** to the subsequent frame"
> > >
> > > > I didn't find evidence for these two claims. While they are quite possible according to the network design, evidence like visualization, case study, or ablation experiments are needed to reveal what the network is really doing. I encourage the authors to provide such evidence in the next revision.
> > >
> > > Thanks for the suggestion. We will add attention visualizations between object and map queries in the appendix.

---

### Review · Reviewer_aE4P · 2025-08-15

**Summary Of Contributions:**

The paper proposes a new structure for a neural architecture for end-to-end autonomous driving called DMAD. Similar to related work, DMAD aims at solving detection, tracking, mapping, prediction, and planning all jointly in a (modular) end-to-end neural architecture. The main goal of the new structure of DMAD is to prevent negative transfer caused (presumably) by conflicting tasks such as object detection and tracking using the same or coupled neural representations. DMAD is designed to separate semantic and motion information more clearly, using a "neural-bayes motion decoder" and an "interactive semantic decoder". The employed motion and object queries share reference point but do not allow gradient flow. This way, coupling between queries is encouraged but feature representations are learned independently. The main contributions are:
 - Identifying negative transfer between semantic and motion learning as a main challenge for modular end-to-end AD learning
 - Proposing the DMAD architecture outlined above to overcome this negative transfer
 - Empirical studies to support the validity and improvements of above's contributions.

**Additional Comments:**

In summary, I think this is a good paper on a neural architecture for modular end-to-end autonomous driving. The proposed approach makes sense and is well motivated, described, and evaluated. It is also of interest to TMLR's audience.

**Audience:**

Yes

**Audience Explanation:**

The paper focuses on a specific application domain of machine learning, namely end-to-end autonomous driving. I expect this paper to be interesting primarily for practitioners and researchers focusing on this domain. While this might only be a specific subgroup of TMLR's audience, I think this group is sufficiently large and the paper of high relevancy for this group. Because of this, I think the paper is clearly above the bar in terms of "relevancy for TMLR".

**Broader Impact Concerns:**

No broader impact concerns form my side.

**Claims And Evidence:**

Yes

**Claims Explanation:**

Overall, the paper provides sufficient empirical evidence to supports the main contributions of the paper:
- SHAP value analysis supports the motivating  hypothesis that semantic and motion information are mixed in the same features (Section 4.5), potentially causing negative transfer.
- DMAD is compared to prior work (UniAD, SparseDrive, PARA-Drive etc.) on the nuScenes dataset. Comparison is done separately for the subtasks of object detection, multi-object tracking, map perception, trajectory prediction, and open- and closed-loop planning (NeuroNCAP). Overall, DMAD (and SparseDMAD) perform favorably, in particular when considering the stage 2 training (training all modules jointly)  in which negative transfer could occur. The negative transfer is much reduced compared to baselines.
- Section 4.4 provides an ablation study which "interpolates" between UniAD and DMAD. This ablation demonstrates the benefits of the individual architectural changes of DMAD (interactive semantic decoder, motion queries, inter-layer reference update and inter-race reference update). DMAD only slightly increase inference time but improves on most metrics.

Empirical results on datasets beyond nuScenes would further strengthen the paper.

The exposition of the method could be improved to ease reproducibility (see below). Ideally, an open-source release of the method would be considered to foster future research on modular end-to-end autonomous driving.

**Requested Changes:**

Requested changes:
 - It seems negative transfer between tasks needs to be understood with respect to capacity constraints of the neural networks. That is: larger networks should be more capable in terms of representing conflicting tasks independently. Could the authors comment on this and how it affects the motivation for the proposed architecture?
 - Section 3 should be more detailed to ease reproducibility. For instance, prediction and tracking updates should be defined more formally. Also, the details of the neural architecture should be outlined. For instance, the authors state "We employ MLPs to extract trajectories from motion queries..." What is the architecture of the MLPs? Similar questions for other layers of the architecture.
 - Clarification: The authors state that the queue length of 3 in the second stage of UniAD degrades performance. But the queue length employed in DMAD is also only 3. Isn't this a contradiction? What was the reason of not going for longer queue lengths?
 - The authors state "minADE is a true positive metric, it does not fully capture the predictive capabilities of the E2E system". It is not clear what a "true positive metric" is - does it refer to being derived from the true-positive-rate? Also, which aspects of the predictive capabilities or an E2E system does it not cover?
 - Limitations of the approach should be discussed in a bit more detail. The last sentence of the conclusion states "However, our approach cannot be applied to E2E methods that are without query propagation mechanism, e.g., VAD (Jiang et al., 2023). Addressing this limitation can be our future work.". This can be elaborated a bit more and also further limitations should be mentioned, e.g. is the approach real-time capable (with inference times of 0.5s)?
 - Could the authors state more clearly what "SparseDMAD" is and how it compares to DMAD?

Further minor change requests:
 - Some notation in Section 3 is not introduced, e.g.  [∆x,∆y,∆z,w,h,l,θ]
 - Some sentences should be rephrased, for instance "Our method _finally_ surpasses UniAD and SparseDrive " or "Addressing this limitation can be our future work." -> "Addressing this limitation remains future work."

---

> ### Author Response · Authors · 2025-09-04
>
> Thank you for your feedback on our work. Below are our responses to the concerns you raised.
>
> > Q1: It seems negative transfer between tasks needs to be understood with respect to capacity constraints of the neural networks. That is: larger networks should be more capable in terms of representing conflicting tasks independently. Could the authors comment on this and how it affects the motivation for the proposed architecture?
>
> We agree that model capacity is a critical factor in mitigating negative transfer. However, simply increasing the capacity can lead to overfitting and performance degradation. We provide empirical evidence: we replace the motion queries and decoder in SparseDMAD with detection queries and decoder, naming it SparseDrive+. The model has almost same #Params as our method but degrades a lot in performance.
>
> | Method            | #$Q_{object}$ | #$Q_{motion}$ | #Params | NDS↑  | AMOTA↑ | Avg. Col.↓ |
> | ----------------- | --------- | --------- | ------- | ----- | ------ | ---------- |
> | SparseDrive+     | 900×2     | 0         | 123.0 M | 0.522 | 0.355  | 0.143      |
> | SparseDMAD (ours) | 900       | 900       | 123.0 M | **0.534** | **0.395**  | **0.093**      |
>
> A better approach to increase capacity is Mixture of Experts (MoE). However, a standard MoE is not directly applicable here. The inherent coupling between tracking and prediction prevents routing inputs to fully independent experts, where the prediction module must know _which_ object's trajectory it is forecasting.
>
> Our Neural-Bayes motion decoder can be conceptualized as a specialized form of MoE tailored for perception and prediction. We design two distinct "experts" for both tasks, which are independent in gradient flow but can exchange essential information.
>
> > Q2: Section 3 should be more detailed to ease reproducibility. For instance, prediction and tracking updates should be defined more formally. Also, the details of the neural architecture should be outlined.
>
> Thanks for the sugguestion. We will formulate tracking and prediction in section 3, and describe the implementation detail of both proposed decoders in the appendix.
>
> > Q3: Clarification: The authors state that the queue length of 3 in the second stage of UniAD degrades performance. But the queue length employed in DMAD is also only 3. Isn't this a contradiction? What was the reason of not going for longer queue lengths?
>
> This is not a contradiction. Our goal is to ensure a fair comparison, which requires keeping the queue length identical for both our method and the baseline , and in both stages of training. We choose a queue length of 3 as it represents a more computationally efficient and practical setting.
>
> > Q4: It is not clear what a "true positive metric" is - does it refer to being derived from the true-positive-rate? Also, which aspects of the predictive capabilities or an E2E system does it not cover?
>
> Thanks for pointing out the ambiguity. Our intended meaning is that minADE can only be computed for true positive detections. Consequently, it cannot penalize detection failures in an E2E system:
> - It ignores erroneous predictions for non-existent objects (false positives)
> - It overlooks the complete failure to predict for missed objects (false negatives)
>
> Therefore, minADE is blind to perception errors and does not fully capture the prediction capability of the E2E system. We will incorporate this clarification in section 4.3.
>
> > Q5: Limitations of the approach should be discussed in a bit more detail ... also further limitations should be mentioned, e.g. is the approach real-time capable (with inference times of 0.5s)?
>
> Regarding the VAD limitation: our method is fundamentally built upon the query propagation mechanism, where the motion query is propagated across timesteps and continuously updated to represent the motion feature of an object. In contrast, architectures like VAD freshly initialize all queries at every timestep, making it impossible to continuously monitor the motion of an object. This lack of query propagation makes our approach incompatible without a major redesign. We will add above elaboration in revision.
>
> Regarding realtime capability: the high latency is not a limitation of our method, but of the heavy UniAD baseline. When our method is applied to the lightweight SparseDrive baseline, the resulting model (SparseDMAD) runs at approximately 10 Hz, confirming its real-time capability.
>
> > Q6: what "SparseDMAD" is and how it compares to DMAD?
>
> Thanks for pointing out the confusion. To be concise, our core contribution is the divide-and-merge structure. The models in our paper are specific applications of this structure:
> - DMAD = our structure + the UniAD baseline
> - SparseDMAD = our structure + the SparseDrive baseline
>
> We will calrify it in the section 4.1.
>
> > Q7: Further minor change requests
>
> Thanks, we will revise accordingly.

---

### Author Response · Authors · 2025-09-11

We thank all the reviewers for their suggestions. We have uploaded the revised manuscript, which incorporates the following changes:

- (Q1.1 of Reviewer pEyw) Section 1, paragraph 1: describe the shared embedding
- Section 1, paragraph 1: add "potentially leading to suboptimal final planning performance"
- (Q1 of Reviewer MLfT) Section 1, bullet 1: clarify the tasks
- Section 3.1, paragraph 2: reformualte object and map queries for the convience of later formulations in updating and tracking.
- (Q7 of Reviewer aE4P) Section 3.1, paragraph 3: add notations to x,y,z,w,h,l ...
- (Q2 of Reviewer aE4P) Section 3.2: formulate updating and tracking
- (Q6 of Reviewer aE4P) Section 4.1, paragraph 1: clarify the naming of DMAD and SparseDMAD
- (Q7 of Reviewer aE4P) Section 4.2, paragraph 2: replace "finally" wiht "ultimately"
- (Q4 of Reviewer aE4P) Section 4.3, paragraph 1: replace "true positive metric" with "can only be computed for true positive detections"
- Section 4.5: swap the "qualitative comparison" and "SHAP value analysis" for a better figure layout.
- (Q5 and Q7 of Reviewer aE4P) Section 5: elaborate the limitation and some minor rephrasing
- (Q3 of Reviewer MLfT) Appendix A
- (Q2 of Reviewer aE4P) Appendix E
- (Q2.2 of Reviewer pEyw, and Q2 of Reviewer MLfT) Appendix H
- (Q2.8 of Reviewer pEyw) Appendix I
- (Q2.8 of Reviewer pEyw) Appendix J

---

### Author Response · Authors · 2025-11-14

We uploaded the camera-ready version with following changes on 10 Nov. 2025.

- Add object-map attention heatmaps in Appendix J
- Improve hyperref appearance: use the same color and remove boxes
- Fix an error in appendix C: the equation should be proportional to
- Fix some grammar issues

---

### Decision · Action_Editor_nSQo · 2025-10-11

**Recommendation:** Accept as is

**Audience:**

Yes

**Audience Explanation:**

Yes. This work addresses a relevant problem in end-to-end autonomous driving and would be of clear interest to the TMLR audience working on multi-task learning, autonomous driving systems, and transformer-based architectures. The paper presents a practical solution with demonstrated performance gains and provides insights into positive versus negative transfer in multi-task scenarios. The method's applicability to multiple baseline architectures (UniAD, SparseDrive) increases its potential impact.

**Claims And Evidence:**

Yes

**Claims Explanation:**

Yes. The paper provides sufficient evidence to support its claims about mitigating negative transfer in end-to-end autonomous driving through the proposed divide-and-merge architecture. The empirical evaluation on nuScenes demonstrates consistent improvements across detection, tracking, prediction, and planning tasks when applied to both UniAD and SparseDrive baselines. The SHAP value analysis provides valuable insights into the negative transfer phenomenon. While one reviewer suggested that evaluation on additional datasets beyond nuScenes would strengthen the evidence, the existing results are convincing for the core claims. The authors adequately addressed reviewer concerns through clarifications and revisions during the discussion period.